# Tackling the Generative learning trilemma through VAE and GMM-controlled latent space class expansion

## Abstract

Achieving efficient data augmentation (DA) in time series classification is not a trivial task due to the high complexity of temporal data. Generative models, such as GANs (Generative Adversarial Networks), diffusion models, and Variational Autoencoders (VAEs), are powerful techniques to address the generative learning trilemma of producing (1) high-quality samples, (2) fast sampling, and (3) diversity. These methods vary in their ability to address the trilemma. Diffusion models allows for high diversity and high quality samples, while GAN allows for high quality samples and fast sampling, and VAE for high diversity and fast sampling. In this paper, we introduce a novel generative method, ASCENSION (V**A**E and GMM-controlled latent **s**pace **c**lass **e**xpa**nsion**), that retains the strengths of VAE in terms of diversity and fast sampling, while enabling controlled and quantifiable exploration of uncharted regions in the latent space. This approach not only enhances classification performance but also yields higher quality (more realistic) samples. ASCENSION leverages the probabilistic nature of the VAE's latent space to represent classes as Gaussian mixture models (GMMs). By modifying this mixture, we enable precise manipulation of class probability densities and boundaries. To ensure intra-class compactness and maximize inter-class separation, we apply clustering constraints. Empirical evaluations on the UCR benchmark (102 datasets) show that ASCENSION outperforms state-of-the-art DA methods, achieving an average classification accuracy improvement of approximately 7% and excelling in all aspects of the generative learning trilemma.

## 1 Introduction

The complexity of time series data, represented as $\mathcal{X} = x_1, x_2, \ldots, x_N$, where each sample $x_i$ belongs to a class $y_i \in 1, 2, \ldots, C$, combined with limited availability of real-world data due to privacy concerns, poses challenges for effective machine learning training. Data augmentation (DA) helps mitigate this issue by generating synthetic data to enhance the training set. DA involves creating an augmented dataset $\mathcal{X}_{\text{aug}}$, which adds new, diverse samples that remain consistent with their respective classes, with the goal of improving the efficiency of the classification model. Formally, let $D_{\text{train}}$ represent the original training dataset and $D_{\text{aug}}$ the augmented dataset. The conventional approach aims to achieve $D_{\text{train}} \cup D_{\text{aug}} \sim d_{\text{true}}$, where $d_{\text{true}}$ denotes the true underlying data distribution.

DA methods fall into two categories: Traditional and Generative DA models/methods (Iglesias et al., 2023b). **Traditional DA methods**, such as AutoAugment (AA) Cubuk et al. (2019) and Fast AutoAugment (FAA) (Lim et al., 2019), automate the application of predefined transformations like window slicing, jittering, or scaling (Iglesias et al., 2023a). However, the reliance on these predefined transformations – *often adapted from the computer vision domain* – restricts the ability to maintain intra-class consistency and preserve the original data semantics, which diminishes the overall effectiveness of the augmentation process. **Generative DA models** like GANs, diffusion models and VAEs (Cheung & Yeung, 2020) address the **generative learning trilemma** of producing (1) high-quality samples, (2) fast sampling, and (3) diversity. While GAN-based DA methods, such as TimeGAN (Zhang et al., 2022), TS-GAN and LatentAugment (Tronchin et al., 2023), excel at generating high-quality samples with speed, they often fall short in terms of diversity (see Figure 1(a)). This limitation arises because these models tend to interpolate within the existing data or introduce

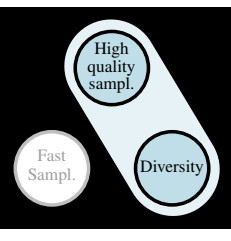 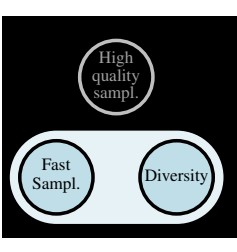 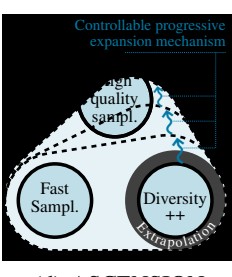

(a) GAN      (b) Diffusion models      (c) VAE      (d) ASCENSION

Figure 1: Overview of how state-of-the-art generative DA models (GANs, Diffusion Models, VAEs) tackle the Generative Learning trilemma versus ASCENSION (the method proposed in this paper). ASCENSION leverages the strengths of VAEs in diversity and fast sampling while enabling controlled, quantifiable exploration – *through data extrapolation* – of uncharted latent space regions, resulting in higher quality samples and improved classification performance.

noise, leading to generated samples that remain confined to the same latent space region as the original data (Xiao et al.). Unlike GANs, diffusion models progressively refine noise into the target data distribution, resulting in highly diverse and high-quality samples. However, they are computationally expensive, making them less efficient than GANs and VAEs for fast sampling (Feng et al., 2024) (see Figure 1(b)). VAEs offer several advantages over GANs and diffusion models. Although GANs achieve fast sampling, VAEs are often even quicker due to their simpler training and generation process. Additionally, the probabilistic nature and structured latent space of VAEs allow for easier control over diversity compared to GANs (see Figure 1(c)). However, to our knowledge, existing methods in the literature (see Appendix A - Related Work) are limited in their capacity to progressively and meaningfully expand class boundaries during synthetic data generation. This limitation presents challenges in situations where the training data distribution does not match the true data distribution, particularly when the training data is collected over a short time frame and does not encompass all potential scenarios encountered during operational phases.

In this research work, we assume that a **controllable progressive expansion mechanism** is crucial to prevent the exploration of regions with a high risk of class overlap, which would degrade sample quality. Despite the advances in state-of-the-art generative DA methods, as outlined in Appendix A and Figure 7, none have ever proposed and integrated such a mechanism into VAEs. To overcome this limitation, we introduce a novel method, ASCENSION, which uses the probabilistic nature of the VAE's latent space to represent classes as Gaussian Mixture Models (GMMs). The core of this approach lies in adjusting the mixture, enabling controlled and measurable exploration of class probability densities and boundaries. This is illustrated in Figure 1(d) where the progressive expansion (shown by successive dashed shapes) reflects different mixture values. Additionally, to ensure that the GMMs faithfully capture the data distribution and retain statistical significance, we impose clustering constraints that enhance the structural properties of the VAE's latent space. These constraints foster intra-class compactness while maximizing inter-class separation.

The main contributions of this paper are:

**C1 (Novel generative DA method for time series)** We introduce ASCENSION, a novel generative method that retains the strengths of VAE in terms of diversity and fast sampling, while enabling controlled and quantifiable exploration of uncharted regions in the latent space. This approach not only improves classification performance but also generates higher-quality samples (*cf.*, Figure 1(d)) through a well-conditioned latent space;

**C2 (Empirical benchmarking on time series data)** We empirically validate ASCENSION's effectiveness and efficiency in addressing the generative learning trilemma and improving classification performance, even in the presence of discrepancies in distance between the training and testing set distributions (cf., Appendix E.2). ASCENSION is benchmarked against both traditional DA methods (FAA) and generative methods (TTS-GAN, LatentAugment, and MODALS);

**C3 (Comprehensive evaluation of conditions enhancing ASCENSION's operational efficiency)** We provide an in-depth analysis of the types of time series – *based on their features* – that are most suitable for augmentation with ASCENSION and the benchmarked methods;

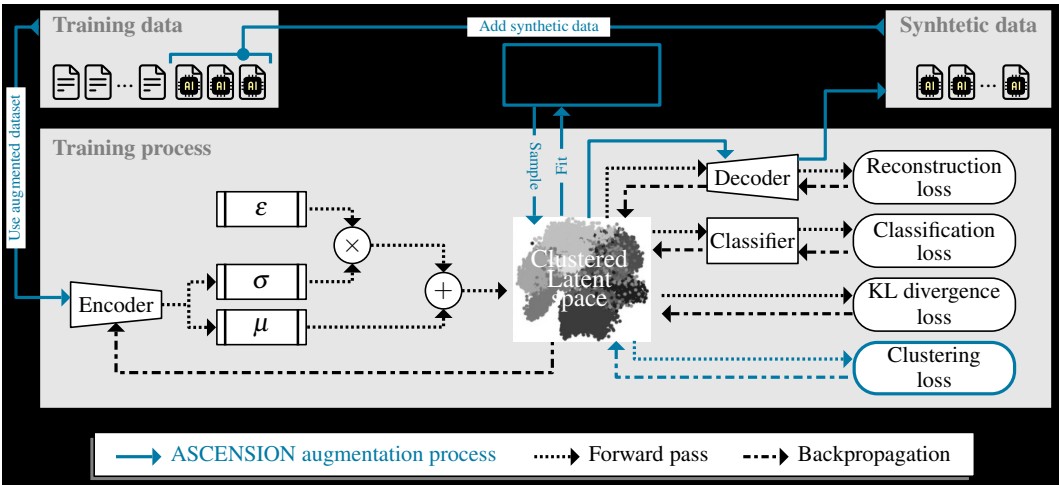

Figure 2: Overview of ASCENSION, leveraging the probabilistic nature of the VAE's latent space to represent classes as GMMs. By modifying this mixture, we enable precise manipulation of class probability densities and boundaries. To ensure intra-class compactness and maximize inter-class separation, clustering constraints are applied. The iterative process enriches, at each round, the training dataset with synthetic samples, resulting in a new dataset $D_{\text{train}_{\text{AUG}}}$ such as: $D_{\text{train INIT}} \subset D_{\text{iter 1}} \subset D_{\text{iter 2}} \subset ... \subset D_{\text{iter N}} = D_{\text{train AUG}}$

## 2 ASCENSION METHOD

ASCENSION builds on a VAE-based generative model, leveraging its fast sampling and diversity capabilities, while introducing a controllable mechanism for progressive expansion of the latent space. Instead of focusing solely on class-consistent augmentation, ASCENSION approximates each class distribution with a GMM. By gradually increasing the variances of GMM components, it expands the space for each class, enabling boundary exploration while minimizing overlap risk. Figure 2 shows ASCENSION's architecture, featuring (i) a deep clustering VAE that learns latent representations, and (ii) a GMM that models the latent space and generates new samples. Encoder and Decoder architectures vary based on data type: fully connected networks for univariate time series, and CNNs or RNNs for multivariate data. The ASCENSION augmentation process involves three steps, outlined in sections 2.1 to 2.3.

### 2.1 VAE TRAINING

The VAE $f_{\text{VAE}}$ to learn a low-dimensional representation of the input time series data. It consists of an encoder $f_{\text{enc}}$ that maps the input data to a latent space, and a decoder $f_{\text{dec}}$ that reconstructs the input data from the latent space. It is worth noting that we implement the re-parametrization trick to differentiate the encoder and decoder during training. In summary: $f_{\text{VAE}} = f_{\text{dec}} \circ f_{\text{enc}} : x \mapsto \hat{x}$, where $\hat{x}$ is the reconstructed version of the input $x$.

### 2.1.1 LATENT SPACE

We denote the latent space as $\mathcal{Z} = \{z_1, z_2, \ldots, z_N\}$, where each latent point $z_i$ corresponds to the encoded vector of the input sample $x_i$. We denote $K$ as the dimension of the latent space. Optimally, $K$ should be chosen as low as possible to capture the essential features of the data while reducing the risk of overfitting. The VAE models the posterior distribution over the latent variables given the input data through the variational distribution $q_\phi(\mathbf{z}|\mathbf{x})$. This distribution is typically assumed to be Gaussian and is parameterized by the encoder network with parameters $\phi$. Specifically, $q_\phi(\mathbf{z}|\mathbf{x})$ is defined as:

$$q_\phi(\mathbf{z}|\mathbf{x}) = \mathcal{N}(\mathbf{z}; \mu_\phi(\mathbf{x}), \Sigma_\phi(\mathbf{x})), \tag{1}$$

where $\mu_\phi(\mathbf{x})$ and $\Sigma_\phi(\mathbf{x})$ represent the mean and covariance of the Gaussian distribution, respectively, both of which are functions of the input $\mathbf{x}$ and are learned by the encoder network.

### 2.1.2 Clustering constraints

**Hypothesis 1 (Latent space clustering)** **[H1]** *We hypothesize that adding a clustering constraints during VAE training will create a more structured latent space, improving class-consistent sample generation and classification performance.*

To ensure that the latent space representations learned by the VAE are semantically meaningful and aligned with the classification task, we introduce clustering constraints during the training process. These constraints encourage samples from the same class to cluster together while maintaining a significant distance from samples of other classes. This approach is vital for generating synthetic samples that are class-consistent and reflect the underlying data distribution. The constraints are incorporated as additional loss terms in the VAE training process, penalizing pairwise distances between samples from the same class in the latent space. By minimizing this clustering loss, the VAE learns to encode the input data in a manner that promotes the generation of diverse and class-specific synthetic samples. In this context of multiple loss terms, normalizing these terms is essential to ensure proper model convergence, preventing any single loss term from dominating the training process. Consequently, we augment the standard VAE loss function with an additional term:

$$\mathcal{L} = \mathcal{L}_{\text{recon}} + \mathcal{L}_{\text{KL}} + \mathcal{L}_{\text{class}} + \mathcal{L}_{\text{cluster}} \tag{2}$$

where $L_{\text{recon}}$ represents the MSE between the VAE's input and output, $L_{\text{KL}}$ denotes the Kullback-Leibler divergence loss, $L_{\text{class}}$ is the classification loss, and $\mathcal{L}_{\text{cluster}}$ is defined as in (3).

$$\mathcal{L}_{\text{cluster}} = \sum_{i=1}^{N} \sum_{j=1}^{N} \delta_{y_i, y_j} \cdot d(z_i, z_j) \tag{3}$$

Given the high dimensionality of the data, we use cosine similarity as the distance metric for $d$. Examples showing how the latent space is evaluating through the learning phase in ASCENSION (using specific UCR datasets) are given and discussed in Appendix F.

## 2.2 GMM manipulation

**Hypothesis 2 (Distribution discrepancies)** **[H2]** *We hypothesize that current state-of-the-art generative DA methods are hindered by significant discrepancies in distance between the training and testing set distributions.*

**Hypothesis 3 (Consistency through expansion)** **[H3]** *We hypothesize that adjusting class distribution to expand training set boundaries will improve accuracy, especially in datasets with discrepancies between training and testing distributions.*

ASCENSION approximates the distribution of each class $y_i$ using a GMM, denoted as $GMM(y_i)$. The augmentation process generates synthetic samples by sampling from these GMMs while gradually expanding the class boundaries by increasing the covariance matrices $\Sigma$ of the Gaussian components. Statistically, samples are generated according to the following formula:

$$x \sim \sum_{k=1}^{K} \pi_k \mathcal{N}(\mu_k, \alpha \Sigma_k) \tag{4}$$

where $\pi_k$ represents the weights of the mixture, $\mathcal{N}(\mu_k, \alpha \Sigma_k)$ is the $k$-th Gaussian distribution component with mean $\mu_k$, and $\alpha \Sigma_k$ is a scaled covariance matrix. The scaling factor $\alpha$ is used to flatten the distribution. Figure 3 illustrates the evolution of class distributions for three different classes as the parameter $\alpha$ changes ($\alpha = 1, 2$ and $3$). As the covariance increases, the overlap between classes may become more pronounced. When significant overlap occurs, the synthetic sample $x$ is assigned to the class $y_i$ with the highest posterior probability, where $P_j(x)$ denotes the posterior probability that sample $x$ belongs to class $y_j$. Formally, the label is chosen as:

$$y_i = \arg\max_{y_j} P_j(x) \tag{5}$$

By carefully controlling the augmentation process, we aim to enrich the training set with synthetic samples that expand decision boundaries while maintaining intra-class consistency and preserving the semantic properties of the original data, ultimately enhancing the model's generalization capabilities. Based on our experiments (see section 3.2.2 and Appendix C), the optimal value for $\alpha$ is found to be slightly above 1, facilitating a gradual exploration of the latent space.

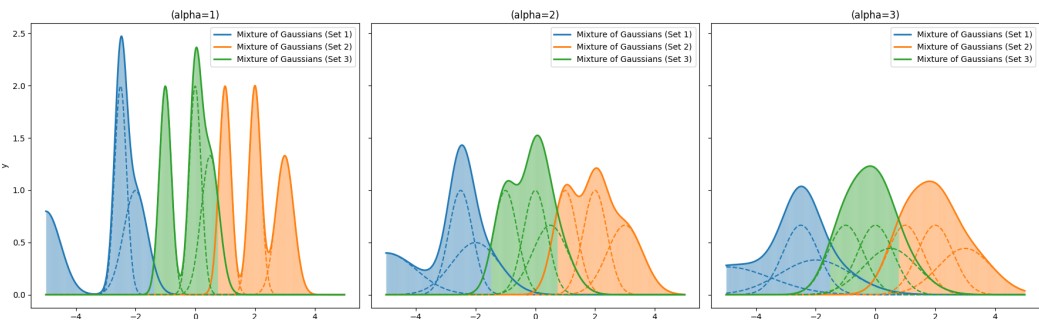

Figure 3: Smoothing process of multi-class GMM. When $\alpha = 1$ the model adheres to the standard GMM definition. As $\alpha$ increases, the model explores less dense regions of the distribution.

## 2.3 SAMPLING, DATA GENERATION & RETRAINING

New samples are generated from this newly fit distribution, decoded, and then incorporated into the base training set. To ensure a clear distinction between classes and to control the overlap between different density functions, a safety measure is implemented. This measure stipulates that if any sampled point from a class has a higher density value in another class, that point is discarded and not added to the training set.

Subsequently, the classifier is retrained using the new samples generated in step 3. This augmentation loop is formally outlined in Algorithm 1. The loop is iteratively repeated until convergence is achieved, resulting in an augmented training dataset that enhances the performance of the classification model.

---

**Algorithm 1:** Augmentation Loop with distinct classes

---

**Input:** Original time series data $\mathbf{X} = \{\mathbf{x}_1, \mathbf{x}_2, \ldots, \mathbf{x}_n\}$ with class labels $\mathbf{Y} = \{y_1, y_2, \ldots, y_n\}$
**Output:** Augmented training dataset $\mathbf{X}_{\text{aug}}, \mathbf{Y}_{\text{aug}}$
**Initialization:**
$\mathbf{X}_{\text{aug}} \leftarrow \mathbf{X}$ ;
$\mathbf{Y}_{\text{aug}} \leftarrow \mathbf{Y}$ ;
**while** *augmentation desired* **do**
    **Train VAE:**
    $\mathcal{L}_{\text{VAE}} = \mathcal{L}_{\text{recon}} + \mathcal{L}_{\text{KL}} + \mathcal{L}_{\text{cluster}} + \mathcal{L}_{\text{class}}$
    $\theta^*, \phi^* \leftarrow \arg\min_{\theta,\phi} \mathcal{L}_{\text{VAE}}$ using $\mathbf{X}, \mathbf{Y}$ ;
    **Fit GMM:**
    Let $\mathbf{Z} = \{\mathbf{z}_1, \mathbf{z}_2, \ldots, \mathbf{z}_n\}$ be the latent representations where $\mathbf{z}_i \sim q_\phi^*(\mathbf{z}|\mathbf{x}_i)$ ;
    Fit a GMM $p(\mathbf{z}|y) = \sum_{k=1}^{K} \pi_k(y)\mathcal{N}(\mathbf{z}|\mu_k(y), \Sigma_k(y))$ to $\mathbf{Z}$ for each class $y$ ;
    **Sample Latent Points:**
    **for** *each class $y$* **do**
        $\mathbf{Z}_{\text{new}}^y = \{\mathbf{z}_1^{\prime y}, \mathbf{z}_2^{\prime y}, \ldots, \mathbf{z}_m^{\prime y}\} \sim p(\mathbf{z}|y)$ ;
        **for** *each $\mathbf{z}_i^{\prime y} \in \mathbf{Z}_{new}^y$* **do**
            **if** *density of $\mathbf{z}_i^{\prime y}$ is higher in another class $y'$* **then**
                Drop $\mathbf{z}_i^{\prime y}$ ;

    **Decode Latent Points:**
    **for** *each class $y$* **do**
        $\mathbf{X}_{\text{syn}}^y = \{\mathbf{x}_1^{\prime y}, \mathbf{x}_2^{\prime y}, \ldots, \mathbf{x}_m^{\prime y}\}$ where $\mathbf{x}_i^{\prime y} = f_\theta^*(\mathbf{z}_i^{\prime y}), \forall \mathbf{z}_i^{\prime y} \in \mathbf{Z}_{\text{new}}^y$ ;
    **Update Training Set:**
    $\mathbf{X}_{\text{aug}} \leftarrow \mathbf{X}_{\text{aug}} \cup \left( \bigcup_y \mathbf{X}_{\text{syn}}^y \right)$ ;
    $\mathbf{Y}_{\text{aug}} \leftarrow \mathbf{Y}_{\text{aug}} \cup \left( \bigcup_y \{y\} \times \mathbf{X}_{\text{syn}}^y \right)$ ;

---

## 3 EXPERIMENTS

### 3.1 EXPERIMENTAL SETUP

**Train/Test datasets:** Experiments were conducted using the UCR Time Series Archive, which comprises 120 univariate time series datasets from various applications and domains, including sensors, ECG, etc. (a complete list of the dataset types is provided in Table 4).

**Classification models:** Classifiers selected for our experiments were chosen based on the findings of Fawaz (2020), which reports that ResNet-50 and Fully Connected Networks (FCN) are the two most effective classifiers (out of 9 evaluated for the UCR datasets. We use the architectures from (Koonce & Koonce, 2021) and (Scabini & Bruno, 2023) for these two classifiers. Additionally, we also incorporate: (i) the embedded classifier of ASCENSION, denoted $\text{ASCENSION}_{\text{EmbCl.}}$; (ii) a combination of ASCENSION's embedded classifier with the state-of-the-art classifiers denoted by $\text{ASCENSION}_{c\text{-EmbCl.}}$ with $c \in \{\text{ResNet, FCN}\}$ in our experiments. The augmentation is defined as the difference between the maximum baseline accuracy (i.e., without augmentation), either VAE's classifier or standalone classifier $c$, and the maximum accuracy achieved by $\text{ASCENSION}_{\text{EmbCl.}}$ or classifier $c$, given by the formula:

$$\text{Acc}_{\text{ASCENSION}_{c\text{-EmbCl.}}} = \max(\text{Acc}_{\text{ASCENSION}_{\text{EmbCl.}}}, \text{Acc}_c) - \max(\text{Acc}_{\text{Baseline}_c}, \text{Acc}_{\text{VAE}}) \quad (6)$$

**Benchmarked DA methods:** ASCENSION is compared with several state-of-the-art methods, including one traditional DA method (FAA) and three generative methods (TTS-GAN, LA, MODALS). More details on these methods can be found in Appendix A. FAA was selected due to its comparable performance with other traditional DA methods (incl., RA and DAA), while MODALS was chosen for its architectural similarity to ASCENSION. TTS-GAN and LA were included as the most recent generative DA methods with publicly available code (*cf.*, Figure 7). However, benchmarking MODALS on the UCR datasets is not feasible since its code, released in 2020, is no longer functional, and the authors informed us they do not plan to repair it. Therefore, we propose to benchmark ASCENSION by evaluating it on the same dataset originally used by Cheung & Yeung (2020) for assessing MODALS.

### 3.2 EXPERIMENTAL RESULTS

#### 3.2.1 PERFORMANCE EVALUATION

**Accuracy:** Appendix B.1 gathers pre- and post-augmentation classification results for the benchmarked techniques, selected classifiers, and UCR datasets. For clarity purposes, Table 1 groups the results in three categories: *(i) Augmented:* refers to the datasets where the performance post-augmentation is better than pre-augmentation; *(ii) Unchanged:* refers to the datasets with no significant improvement or degradation ($\pm 10^{-4}\%$) of performance post-augmentation, *(iii) Worsened:* refers to the datasets where the augmentation of the train set degrades performance. Under each category we report the number of datasets and mean accuracy post-augmentation for the different configurations (classifiers, DA methods).

Several findings can be drawn from Table 1. First, while FAA shows a mean improvement of $5.12\%$ (ResNet) and $5.68\%$ (FCN), it does not generalize well, as it only improves accuracy on $18/102$ datasets (ResNet) and $24/102$ (FCN). In contrast, $\text{ASCENSION}_{\text{ResNet-Emb}}$ improves accuracy on $68$ datasets (ResNet) and $64/102$ (FCN), with mean accuracy gains of $3.97$ and $2.08\%$, respectively. The slightly lower mean improvement for ASCENSION and ASCENSION$c$-Emb compared to FAA is due to the larger number of datasets successfully augmented, including those with smaller, yet positive, improvements, as detailed in Appendix B.1 . ASCENSION and $\text{ASCENSION}_{c\text{-Emb}}$ augment more than twice as many datasets as the benchmark methods (FAA, LA, TTS-GAN), highlighting the superior generalization ability of ASCENSION and supporting **[H1]**. Finally, when compared to MODALS on the HAR dataset (Table 2), ASCENSION further enhances performance. While MODALS improves the baseline classification (without augmentation) by $3.23\%$, ASCENSION increases this improvement by $+4.78\%$, further advancing accuracy beyond the baseline.

**Trilemma performance:** Table 3 presents an analysis of how the benchmarked methods perform across each aspect of the generative learning trilemma. ASCENSION stands out with impressive results, particularly in sample quality, which rivals that of a GAN-based approach, while showing

Table 1: Results of our empirical benchmark study on the 102 UCR datasets. The table summarizes the number of datasets with improvements (Augmented), no change (Unchanged), and deterioration (Worsened) in accuracy for each method. The mean accuracy change ($\overline{Acc}$) is provided for each category. An upward arrow (↑) indicates that higher values are preferable, while a downward arrow (↓) signifies that lower values are advantageous. Bold values denote the best performance, and underlined values indicate the second best. ASCENSION achieves the highest number of improved datasets and the fewest cases of worsened performance, demonstrating its effectiveness in enhancing classification accuracy across the datasets.

| | DA method | Augmented | | Unchanged | | Worsened | | ↑Total | |
|---|---|---|---|---|---|---|---|---|---|
| | | ↑Nb$_{datasets}$ | ↑$\overline{Acc}$ | Nb$_{datasets}$ | $\overline{Acc}$ | ↓Nb$_{datasets}$ | ↑$\overline{Acc}$ | Nb$_{datasets}$ | ↑$\overline{Acc}$ |
| ResNet | FAA | 18 | **5.12%** | 13 | **0%** | 71 | -8.54% | 102 | -4.59% |
| | LA | 14 | 1.03% | 11 | **0%** | 77 | -5.54% | 102 | -4.04% |
| | TTS-GAN | 24 | 3.07% | 9 | **0%** | 69 | -7.08% | 102 | -4.17% |
| | ASCENSION | 52 | 3.01% | 14 | **0%** | 36 | -1.55% | 102 | 0.99% |
| | ASCENSION$_{ResNet-Emb}$ | **68** | 3.97% | 15 | **0%** | **19** | **-1.06%** | 102 | **2.45%** |
| FCN | FAA | 23 | **5.68%** | 10 | **0%** | 69 | -8.44% | 102 | -4.44% |
| | LA | 20 | 3.69% | 14 | **0%** | 68 | -3.58% | 103 | -1.54% |
| | TS-GAN | 24 | 1.54% | 15 | **0%** | 57 | -9.24% | 102 | -5.07% |
| | ASCENSION | 60 | 2.72% | 17 | **0%** | 26 | **-1.66%** | 104 | -1.16% |
| | ASCENSION$_{FCN-Emb.}$ | **64** | 2.08% | 14 | **0%** | **25** | -1.68% | 103 | **-0.89%** |
| | ASCENSION$_{Emb.}$ | 51 | 1.93% | 22 | **0%** | 29 | *-1.72%* | 102 | *0.48%* |

Table 2: Acc. comparison on HAR dataset used by (Cheung & Yeung, 2020) to assess MODALS

| Method | Accuracy (%) |
|---|---|
| ASCENSION$_{ResNet-Emb}$ | **93.42** |
| MODALS | 91.87 |
| No Augmentation | 88.64 |

a notable improvement in sample diversity (largely due to the expansion of Gaussian mixtures). Additionally, these strong performances are achieved without increasing computational cost, as AS-CENSION's sampling speed matches that of TTS-GAN and is more than three times faster than FAA. For more detailed results, refer to Appendix B.2, where it is shown that ASCENSION consistently delivers stable outcomes across all UCR datasets, in terms of both quality and diversity, with a clear trend of outperforming TTS-GAN and FAA.

Table 3: Comparison of Metrics for Different Methods - Mean metrics over a subset of 11 datasets from UCR archive, one from each domain to ensure representativity despite computational costs. The metrics are defined in Appendix E.1

| Metric | ASCENSION | | TTS-GAN | | FAA | |
|---|---|---|---|---|---|---|
| | mean | median | mean | median | mean | median |
| Quality | **1.01** | **1.00** | 0.99 | 0.99 | 0.99 | 1.00 |
| Diversity | $\mathbf{1.690 \times 10^{10}}$ | **1538.55** | $1.43 \times 10^7$ | 1188.68 | $6.20 \times 10^8$ | 57.18 |
| Fast sampling (Speed) | **0.2** | | **0.2** | | 0.9 | |

### 3.2.2 HYPERPARAMETERS SENSITIVITY ANALYSIS

A key feature of ASCENSION is its controllable progressive expansion mechanism for exploring the latent space. Adjusting the scaling factor parameter $\alpha$ (which influences how distributions are flattened, see section 2.1) and determining the number of iterations are essential for optimizing the method's effectiveness. These two parameters must be carefully balanced to maintain sufficient separation between distributions while allowing for adequate exploration. Both excessive and in-

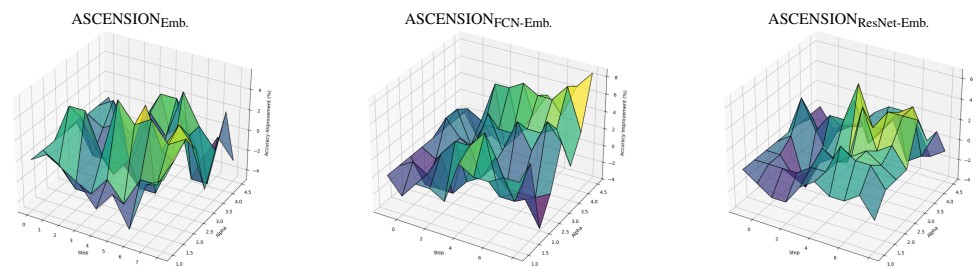

Figure 4: Analysis of accuracy augmentation as a function of the parameter $\alpha$ and the number of augmentation steps for the **Ham** dataset. The results suggest that clearly defining optimal values for $\alpha$ and the maximum number of iterations is challenging. However, it is evident that $\alpha$ should remain above 1, and a minimum threshold of approximately 3 iterations is deemed acceptable. A comprehensive grid search may be warranted to identify the optimal parameter values. More examples can be found in appendix C.

sufficient overlap between distributions can negatively affect the accuracy and overall performance of the generated outputs (see Appendix F for visualizations of how the exploration evolves over iterations for several UCR datasets). However, one could argue that if newly generated data are discarded when the density of another class exceeds that of the current labeled class, the significance of $\alpha$ diminishes, as a safeguard is already in place.

**Analysis methodology:** We conducted a study that varied $\alpha$ (from 1 to 5) and the number of iterations (from 1 to 9) to assess their impact on accuracy improvement and determine whether convergence occurs.

**Results:** Figure 4 presents the results for ASCENSION$_{\text{EmbCl.}}$, ASCENSION$_{\text{ResNet-EmbCl.}}$, and ASCENSION$_{\text{FCN-EmbCl.}}$ using the **Ham** dataset from the UCR archive (additional examples can be found in Appendix C). The augmentation process remains relatively stable even with high $\alpha$ values, supporting our hypothesis that the distribution borders reduce the sensitivity of $\alpha$ in this method. Appendix C offers similar analyses across various UCR datasets, showing that increasing $\alpha$ can enhance boundary exploration but may reduce performance if $\alpha$ is too large. Based on our experiments, selecting $\alpha$ in the range $[1, 3]$ provides a good balance.

### 3.2.3 ANALYSIS OF CONDITIONS ENHANCING ASCENSION'S OPERATIONAL EFFICIENCY

Section 3.2 has empirically evidenced that, for the majority of applications (datasets), ASCENSION outperforms traditional and generative state-of-the-art methods. However, there remains a significant portion of datasets (approximately 30% to 50%) where ASCENSION does not improve classification performance and in some cases, even worsens it (refer to the results in the Unchanged and Worsened columns in Table 1). Reader can refer to Appendix B.1 to have a complete overview of which datasets remain unchanged or are degraded. Therefore, we propose conducting an analysis to identify the types of data – *based on their features* – that benefit the most from augmentation and those that require minimal augmentation.

**Feature extraction:** We use the CATCH22 time series feature set introduced by Lubba et al. (2019) to characterize the datasets (comprising 22 features in total), adding the ratio of train/test split and the distribution discrepancy ratio between train and test (cf., Appendix E.2). A description of these 24 features (F1-F24) is provided in Appendix D.

**Analysis methodology:** By averaging the features of the time series in each dataset, we identify the datasets that are most and least amenable to benefit from augmentation. Subsequently, we analyze the impact of augmentation on the classification performance of these datasets to determine the most influential features. To measure feature importance, we employ a random forest model with a high number of estimators with low depth to the mean of F1-F24 to predict augmentation for the benchmarked DA methods.

**Results:** Figure 5 shows that F2 is among the most critical features in the time series dataset, influencing all DA methods (either enhancing or diminishing classification performance). Additionally, we observe that each method is strongly tied to specific features: FAA to F10 (which gauges the

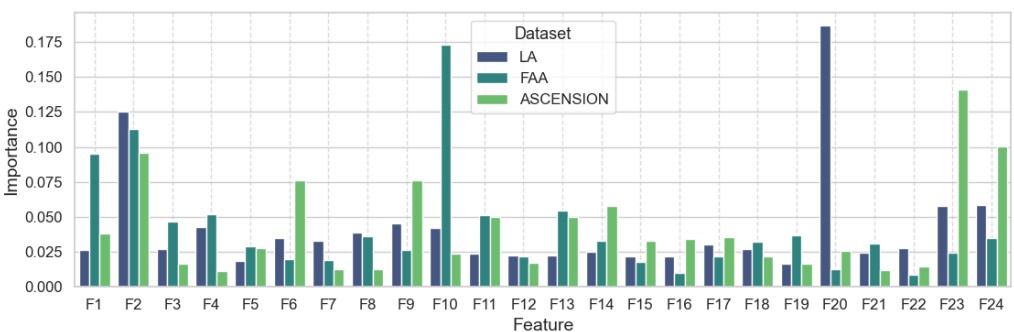

Figure 5: Feature importance derived from a random forest model applied to the 24 features (F1-F24, cf. Appendix D.). F10 (to what extent a pattern is repetitive in a time series), F20 (part or fraction of fluctuations that occur over longer periods of time), F23 (ratio of train and test data in the dataset), F24 (discrepancy in distance between the train and test set distributions, see Appendix E.2).

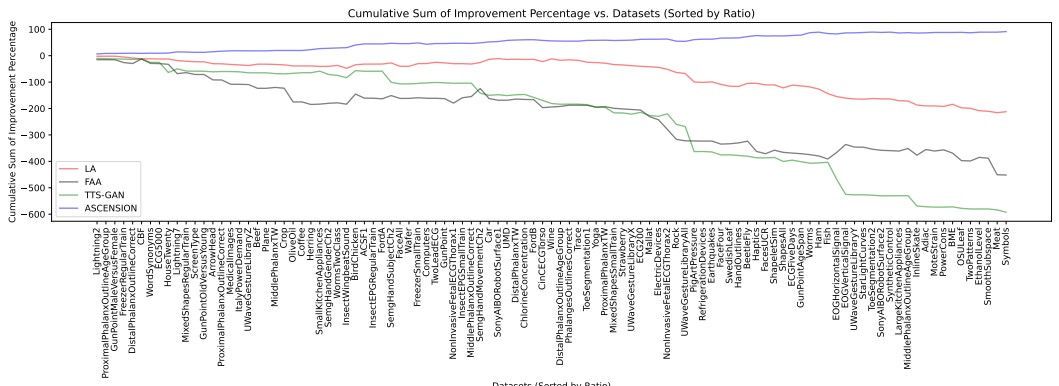

Figure 6: Cumulative sum of classification performance improvements as a function of dataset discrepancies between train and test sets (see Appendix E.2). Datasets are ordered according to their discrepancy values.

degree of periodic patterns within the dataset), LA to F20 (which reflects fluctuations over extended periods), and ASCENSION to F23 and F24 (respectvely representing the train/test ratio of data and discrepancy in distance between the training and testing set distributions, *cf.* Appendix E.2). This last finding aligns with our expectation that ASCENSION takes special care of exploring the latent space more thoroughly, thus being closely linked to the distributional differences between the training and test datasets.

To further analyze how the classification performance for the benchmarked DA methods evolves along with the increase in discrepancy in distance between the training and testing sets, we plot in Figure 6 the cumulative sum of classification performance improvements (%) as a function of F24 (see Appendix E.2 (the 102 UCR datasets on the x-axis have been ordered from the smallest to the highest discrepancy). It can be observed that, while other DA methods tend to result in lower performance as the discrepancy ratio increases, ASCENSION maintains positive performance, and even shows a slight increase. This validates our hypothesis **[H2]**, which assumed that existing (state-of-the-art) DA methods are unable to tackle datasets facing discrepancy situations, but also **[H3]** that assumed that empowering DA methods with the ability to explore previously uncharted regions in the train/test space can lead to enhanced classification performance.

## 4 CONCLUSION

A key challenge in time series data augmentation is addressing the generative learning trilemma (see Figure 1). Generative DA methods, such as GAN, diffusion models and VAEs vary in their ability to address this trilemma while maintaining high classification performance. In this paper, we introduce a novel method called ASCENSION, which builds on the strengths of VAEs in terms of diversity and fast sampling, while enabling controlled and quantifiable exploration of uncharted regions in the latent space. ASCENSION uses the probabilistic nature of the VAE's latent space to model classes as GMMs, and through the manipulation of these mixtures, allows for precise adjustments to class probability densities and boundaries. Clustering constraints are applied to maintain intra-class compactness and maximize inter-class separation. Overall, ASCENSION addresses the challenges of high-dimensional sequential data by enhancing the representativeness of the training set and expanding decision boundaries in a controlled manner. This is particularly useful when there is a significant discrepancy in distance between the distributions of training and testing sets.

Our empirical study on 102 UCR benchmark datasets shows that ASCENSION outperforms state-of-the-art DA techniques. It is evaluated on multiple metrics, including (i) classification accuracy, (ii) diversity, (iii) sample quality, and (iv) fast sampling speed, excelling in all areas compared to benchmarked DA methods. Furthermore, an in-depth analysis identifies the types of time series data that benefit most from augmentation with each DA method. This study highlights ASCENSION's advantage in handling datasets with high discrepancies between training and testing distributions.

Future research could explore extending ASCENSION to other types of sequential data, such as natural language or spatio-temporal datasets, but also non-sequential data such as images, due to its highly flexible architecture. We could also explore new clustering and sampling strategies to enhance generalization across different domains, along with expansion mechanisms (e.g., beyond a single $\alpha$ factor).

## 5 REPRODUCIBILITY

The UCR time series archive can be found at `https://www.cs.ucr.edu/~7Eeamonn/time_series_data_2018/`. We detailed exact implementation details and provide code to produce our results on an anonymous github page at `https://github.com/ASCENSION-PAPER`

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

## A  RELATED WORK

Iglesias et al. (2023b) and Iwana & Uchida (2021) divide DA for time series into two categories: Traditional *vs.* Generative DA methods. Figure 7 offers an overview of the evolution of these methods, emphasizing their ability to address and manage the generative learning trilemma (diversity, high-quality samples, fast sampling) and whether their associated codes are publicly available.

**Traditional DA methods**, such as window slicing, jittering, and scaling (Iglesias et al., 2023a), are primarily adapted from computer vision and rely on transformation strategies like cropping, rotation, scaling, drifting, and so forth. However, the complex nature of time series data often renders these methods sub-optimal, as they can disrupt the semantic integrity of the original data. For instance, while a slightly flipped image of a cat remains recognizable, reversing the time axis of an electrocardiogram sequence can render it meaningless. In response to these challenges, more advanced DA techniques were developed to automate the sequence of transformations to be performed. A first method, named **AutoAugment (AA)** Cubuk et al. (2019), uses reinforcement learning to explore transformation pipelines/policies. A second method named **Fast AutoAugment (FAA)** (Lim et al., 2019) uses density matching for a faster search strategy, eliminating the need for back-propagation. Subsequent methods such as **RandAugment** (Cubuk et al., 2020), **Deep AutoAugment** (Zheng et al., 2022), and **Trivial Augment** (Müller & Hutter, 2021) were introduced to further simplify and refine the augmentation search strategy. RandAugment streamlines the augmentation process by removing the exhaustive search phase, instead applying a fixed number of random transformations with adjustable magnitudes. Deep AutoAugment incorporates a deep reinforcement learning model that dynamically combines transformation policies based on the specific characteristics of the dataset. Trivial Augment introduces an even simpler approach by applying a minimal set of random transformations, emphasizing ease of use and computational efficiency. Despite all these advancements, all these methods rely on predefined transformations, which is suboptimal for preserving intra-class consistency and the semantic characteristics of the original time series data, thereby limiting the effectiveness of data augmentation.

**Generative DA models** such as Generative Adversarial Networks (GANs) (Goodfellow et al., 2020), diffusion models (Yang et al., 2023a), and VAEs (Kingma & Welling, 2013) represent powerful techniques capable of learning a probabilistic representation of data distributions. These models

Figure 7: Overview of the evolution of state-of-the-art data augmentation methods for time series (traditional vs. generative), highlighting their capacity to address and control the generative learning trilemma: (1) Diversity, (2) High-quality samples, and (3) Fast sampling (Xiao et al.). The symbols $\bigcirc$, $\bullet$, and $\bullet$indicate the degree to which each method addresses and manages these dimensions of the trilemma (ranging from no consideration to full consideration). **MODALS: Although code was made available (4 years ago), it is currently non-functional; we have contacted the authors of MODALS Cheung & Yeung (2020) for the source code, but they informed us that it is no longer operational and cannot be repaired without substantial re-coding.

can generate time series data that retain the temporal dependencies, semantic consistency, and class-specific characteristics of the original datasets Fu et al. (2020). For example, using a representation layer, as introduced by (Liu et al., 2022), provides an abstraction that is crucial when dealing with time series data. **TimeGAN** (Zhang et al., 2022) has been specifically designed for time series, which has shown significant improvements in generating high-quality synthetic sequences and augmenting low-quality datasets. Likewise, **TS-GAN** (Yang et al., 2023b) develop a LSTM-based GAN architecture with an sequential-squeeze-and-excitation to better capture time-dependence between the current and past moments in each dimensions. TS-GAN is particulary proposed to generate augmented sensor-based health data to improve Deep Learning (DL) classification models and evaluated on 3 health time series datasets. **TTS-GAN** (Li et al., 2022) adapt the traditional GAN architecture using a transfomer-encoder architecture that can deal with long range dependencies in time sequences. It shows strong performance in generating realistic data across three datasets: a simulated dataset, a human acuity recognition dataset, and an ECG dataset. However, GANs training process is very unstable and is very senstive to hyperparameters. It also suffers from issue as mode collapse that can limit the variety of generated samples and can possibly generate unrealistic data (Lei et al., 2019). **LatentAugment** (Tronchin et al., 2023) learns a low-level representation of initial data, noising around learned points and then decoding them to produce newly generated and semantically close data. More recently, (Seon et al., 2024) proposed **LISGAN**, a GAN-based architecture to augment time series data in the context of class imbalance by adjusting the loss with mutual information term and using a spectral normalization. LISGAN generates high quality synthetic data and significantly increases classification performance with industrial internet of things datasets. Diffusion models, a more recent class of generative models, have garnered significant attention for their capability to model complex data distributions. Unlike GANs, which rely on adversarial training, diffusion models generate data by progressively refining noise toward the target data distribution. This denoising approach has yielded remarkable results in high-fidelity image generation, as seen with models like DALL·E 2, Imagen, and Flux. Recently, starting in 2023, several diffusion model-based DA methods for time series have emerged, including **ASE-DDPM** Liu et al. (2024) for addressing imbalanced time series classification, **DiffRUL** Wang et al. (2024) for enhancing remaining useful life predictions, **D3A-TS** Solis-Martin et al. (2023) aimed at improving synthetic sample quality through meta-attribute conditioning, and **Time-DDPM**, which integrates a diffusion denoising probabilistic model with CNN-LSTM networks to enhance sample quality. While diffusion models provide stable outputs, they face challenges with long-range predictions, error accumulation, and slow inference (Feng et al., 2024), which can limit their practical applications. VAEs offer several advantages over GANs and diffusion models. Their probabilistic nature allows for explicit control over the diversity and quality of generated samples through manipulation of the

Table 4: UCR dataset types along with the selected representative datasets

| Type | Representative dataset | Description |
|---|---|---|
| Device | ACSF1 | Measurements of alternating current signals for predictive maintenance |
| ECG | ECG200 | Electrocardiogram (ECG) readings used to detect heart abnormalities |
| EOG | EOGVerticalSignal | Electrooculography (EOG) signals capturing eye movement patterns |
| Image | BeetleFly | Shape-based image classification of beetle and fly outlines |
| Motion | Worms | Motion sensor data capturing worm movements for classification |
| Sensor | Car | Sensor readings collected from a car, used for detecting driving conditions |
| Simulated | UMD | Simulated control processes data |
| Spectro | Ham | Spectroscopy data to identify types of ham based on chemical properties |
| Spectrum | SemgHandMovementCh2 | Electromyography (EMG) data of hand movements, recorded across channels |

latent space, as evidenced in (Cheung & Yeung, 2020). This helps preserve the intra-class consistency and semantic characteristics of the original data. Additionally, VAEs are less prone to collapse compared to GANs and are less computationally expensive than both GANs and diffusion models (Thanh-Tung & Tran, 2020). To ou knowledge, the first VAE-based DA model, named **MODALS**, was introduced in (Cheung & Yeung, 2020) and represents the closest architectural approach to AS-CENSION. It was the first study to investigate the expansion of class boundaries during synthetic data generation, although it does not offer a method for controlling this expansion. Recently, Dang et al. (2024) introduced **VAE-LSTM**, which is used to augment an inertial sensor dataset due to limited data availability, with the goal of enhancing classification performance. However, this approach does not explore the expansion of class representations in the latent space, as proposed in ASCENSION.

To our knowledge, none of the aforementioned methods have explored a controllable progressive expansion strategy, which is anticipated – *and demonstrated in section 3.2* – to enhance classification performance and produce higher quality samples. While MODALS has examined an expansion strategy, it lacks control. Results shown in Table 2 indicate that ASCENSION outperforms MODALS when evaluated on the same dataset originally used in Cheung & Yeung (2020), achieving an accuracy of 93.24% compared to 91.87% for MODALS[1].

## B  ENLARGED EXPERIMENTAL RESULT ANALYSIS

### B.1  ENLARGED CLASSIFICATION PERFORMANCE

This section offers a more comprehensive analysis of the results. The 102 datasets from the UCR time series classification repository are grouped into 9 distinct categories (domains/applications), as summarized in Table 4.

A detailed breakdown of our experimental results is presented in Table 5. These results are the ones obtained with ResNet[2], and are aggregated per dataset category (e.g., Device, ECG200, etc., see Table 4). ASCENSION achieves the highest number of improved datasets across nearly all categories (7 out of 8 dataset types). For further details, including accuracy differences before and after augmentation for each dataset and method, refer to Table 6.

---

[1]Although MODALS code was made available in 2020, it is currently non-functional. We have contacted the authors of MODALS Cheung & Yeung (2020) for the source code, they informed us that it is no longer operational and cannot be repaired without substantial re-coding.

[2]ResNet was chosen for this analysis due to its superior average performance (see Table **??**). For more comprehensive results, including those for FCN, visit: https://github.com/ASCENSION-PAPER

Table 5: Mean Improvement per Dataset Type

| Type | FAA | | LA | | TTS-GAN | | ASCENSION$_{ResNet-Emb}$ | |
|---|---|---|---|---|---|---|---|---|
| | ↑Nb$_{augmented}$ | ↑$\overline{Acc}$ | ↑Nb$_{augmented}$ | ↑$\overline{Acc}$ | ↑Nb$_{augmented}$ | ↑$\overline{Acc}$ | ↑Nb$_{augmented}$ | ↑$\overline{Acc}$ |
| Device | 1/8 | 1.06% | 2/8 | 0.01% | 3/8 | 1.51% | **5/8** | **2.15%** |
| ECG | 0/6 | 0.0% | 2/6 | 5.20% | 4/6 | **5.63%** | **5/6** | 1.80% |
| EOG | **2/2** | **6.21%** | 0/2 | 0.0% | 0/2 | 0.0% | 1/2 | 3.86% |
| Image | 13/32 | 5.41% | 5/32 | 6.31% | 7/32 | 3.71% | **21/32** | **6.73%** |
| Motion | 1/14 | 1.29% | 2/14 | 2.14% | 0/14 | 0.0% | **8/14** | **2.71%** |
| Sensor | 2/20 | 1.03% | 6/20 | **4.70%** | 5/16 | 2.63% | **12/20** | 2.02% |
| Simulated | 2/8 | 12.6% | 1/8 | **5.33%** | 3/8 | 1.11% | **5/8** | 1.24% |
| Spectro | 2/8 | 3.02% | 2/8 | **5.36%** | 2/8 | 2.26% | **3/8** | 1.58% |

## B.2 ENLARGED GENERATIVE LEARNING TRILEMMA PERFORMANCE

Tables 7 and 8 provide a detailed breakdown of our experimental results regarding the diversity and quality dimensions of the generative learning trilemma (refer to Table 3 for results on the fast sampling dimension).

ASCENSION consistently demonstrates the highest diversity across all dataset types. While this outcome was expected in comparison to TTS-GAN, it was less certain against FAA, as time series transformations can yield highly diverse semantic results in terms of distance from the original data. This diversity stems from expanding the class distribution, enabling our synthetic samples to be drawn from outside the training data distribution in a way that better approximates the real data distribution. This approach helps the generated samples get closer to the unseen testing data, which is treated as the real data, rather than merely reproducing the training data. As evidenced in Table 8, this improved diversity also enhances the quality of the samples, resulting in more realistic synthetic data across all datasets.

Table 6: Difference in accuracy between post- and pre-augmentation with TTS-GAN, FAA, LA and ASCENSION$_{\text{ResNet-Emb}}$. Best improvement in bold.

| Dataset | TTS-GAN | FAA | LA | ASCENSION$_{\text{ResNet-Emb}}$ |
|---|---|---|---|---|
| ACSF1 | $-2.0\%$ | $-14.0\%$ | $2.0\%$ | **4.16**% |
| Adiac | $-2.3\%$ | **13.04**% | $-2.3\%$ | $0.77\%$ |
| ArrowHead | $-1.71\%$ | $-17.14\%$ | $-5.71\%$ | **2.29**% |
| BME | $0.67\%$ | $-9.33\%$ | **5.33**% | $0.01\%$ |
| Beef | $0.0\%$ | $-10.0\%$ | **3.33**% | $0.0\%$ |
| BeetleFly | $-5.0\%$ | $5.0\%$ | **10.0**% | $5.0\%$ |
| BirdChicken | $20.0\%$ | **25.0**% | $10.0\%$ | $10.0\%$ |
| CBF | $1.0\%$ | **14.67**% | $-3.0\%$ | $-0.55\%$ |
| Car | $-5.0\%$ | $-35.0\%$ | **6.67**% | $3.33\%$ |
| ChlorineConcentration | **0.96**% | $-0.78\%$ | $-0.81\%$ | $0.94\%$ |
| CinCECGTorso | $-18.99\%$ | $-24.64\%$ | **−7.32**% | $-2.68\%$ |
| Coffee | **3.57**% | $0.0\%$ | $0.0\%$ | $0.0\%$ |
| Computers | $0.0\%$ | $-1.6\%$ | **0.4**% | $-5.2\%$ |
| Crop | **−0.68**% | $-2.17\%$ | $-1.09\%$ | $0.01\%$ |
| DistalPhalanxOutlineAgeGroup | $-2.16\%$ | **1.44**% | $-4.32\%$ | $-0.72\%$ |
| DistalPhalanxOutlineCorrect | $-2.17\%$ | $-2.9\%$ | $-2.9\%$ | **0.36**% |
| DistalPhalanxTW | $2.16\%$ | **2.88**% | $0.72\%$ | $1.44\%$ |
| ECG200 | **6.0**% | $-2.0\%$ | $-2.0\%$ | $3.0\%$ |
| ECG5000 | $-0.27\%$ | $-0.56\%$ | $-0.67\%$ | **−0.18**% |
| ECGFiveDays | $3.83\%$ | $-2.67\%$ | **8.25**% | $1.74\%$ |
| EOGHorizontalSignal | $-32.32\%$ | **7.46**% | $-5.52\%$ | $-1.38\%$ |
| EOGVerticalSignal | $-19.61\%$ | **4.97**% | $-2.21\%$ | $3.87\%$ |
| Earthquakes | $-1.44\%$ | $0.0\%$ | **1.44**% | $0.1\%$ |
| ElectricDevices | $-1.96\%$ | $-8.82\%$ | $-1.63\%$ | **0.19**% |
| EthanolLevel | $0.0\%$ | **4.2**% | $-3.6\%$ | $2.31\%$ |
| FaceAll | $-5.27\%$ | $-9.88\%$ | $-11.78\%$ | **−1.25**% |
| FaceFour | $-9.09\%$ | $-10.23\%$ | $-7.95\%$ | **4.06**% |
| FacesUCR | **−0.1**% | $-8.0\%$ | $-5.07\%$ | $-1.76\%$ |
| Fish | **1.71**% | $-10.86\%$ | $-12.57\%$ | $-5.35\%$ |
| FordA | $-0.08\%$ | $-2.27\%$ | $0.15\%$ | **0.23**% |
| FordB | $0.0\%$ | $-0.62\%$ | **0.49**% | $0.25\%$ |
| FreezerRegularTrain | $0.0\%$ | $-10.46\%$ | $-3.12\%$ | **0.35**% |
| FreezerSmallTrain | $0.0\%$ | $1.93\%$ | $8.35\%$ | **2.88**% |
| GunPoint | $0.0\%$ | $-1.33\%$ | $-2.0\%$ | **0.0**% |
| GunPointAgeSpan | $0.0\%$ | $-2.22\%$ | $-2.85\%$ | **1.58**% |
| GunPointMaleVersusFemale | **0.0**% | $-0.32\%$ | **0.0**% | **0.0**% |
| GunPointOldVersusYoung | $-0.32\%$ | **0.0**% | **0.0**% | **0.0**% |
| Ham | $0.95\%$ | $-3.81\%$ | $-5.71\%$ | **1.9**% |
| HandOutlines | $0.0\%$ | **2.16**% | $-0.81\%$ | $0.54\%$ |
| Haptics | $-2.92\%$ | $-19.48\%$ | $0.0\%$ | **3.9**% |
| Herring | **0.0**% | $-6.25\%$ | $0.0\%$ | **3.12**% |
| HouseTwenty | $-36.13\%$ | $-2.52\%$ | $0.0\%$ | **0.84**% |
| InlineSkate | $-15.64\%$ | $-6.18\%$ | $-6.0\%$ | **−1.64**% |
| InsectEPGRegularTrain | **0.0**% | **0.0**% | **0.0**% | $0.0\%$ |
| InsectEPGSmallTrain | $0.0\%$ | **16.87**% | $0.0\%$ | $0.0\%$ |
| InsectWingbeatSound | $-4.6\%$ | $-2.17\%$ | $-5.51\%$ | **1.16**% |
| ItalyPowerDemand | $-0.97\%$ | $-0.29\%$ | $-1.55\%$ | **0.1**% |
| LargeKitchenAppliances | **0.27**% | $-1.33\%$ | $-5.6\%$ | $-3.2\%$ |
| Lightning2 | $-8.2\%$ | $-11.48\%$ | $-1.64\%$ | **6.56**% |
| Lightning7 | **9.59**% | $-30.14\%$ | $-4.11\%$ | $4.4\%$ |
| Mallat | $-11.64\%$ | $-23.97\%$ | $-1.54\%$ | **0.34**% |
| Meat | $-3.33\%$ | $-58.33\%$ | $-5.0\%$ | **0.0**% |
| MedicalImages | $-0.13\%$ | $-11.84\%$ | $-1.84\%$ | **1.45**% |
| MiddlePhalanxOutlineAgeGroup | $0.0\%$ | **5.84**% | $-0.65\%$ | $1.3\%$ |
| MiddlePhalanxOutlineCorrect | $0.34\%$ | **3.44**% | $-1.03\%$ | $-0.69\%$ |
| MiddlePhalanxTW | $-1.95\%$ | **1.95**% | $-0.65\%$ | $1.3\%$ |
| MixedShapesRegularTrain | $-7.88\%$ | **3.46**% | $-2.23\%$ | $-0.68\%$ |
| MixedShapesSmallTrain | $-18.56\%$ | $-5.32\%$ | $-4.62\%$ | **−1.15**% |
| MoteStrain | $-1.28\%$ | $-5.11\%$ | $-0.16\%$ | **1.6**% |
| NonInvasiveFetalECGThorax1 | $-2.85\%$ | $-12.72\%$ | $-2.6\%$ | **0.92**% |
| NonInvasiveFetalECGThorax2 | **8.04**% | $-26.51\%$ | $-6.36\%$ | $0.56\%$ |
| OSULeaf | $-4.13\%$ | $-27.69\%$ | $-10.33\%$ | **0.41**% |

| Dataset | TTS-GAN | FAA | LA | ASCENSION$_{\text{ResNet-Emb}}$ |
|---|---|---|---|---|
| OliveOil | 0.0% | −43.33% | −3.33% | **0.2**% |
| PhalangesOutlinesCorrect | 0.23% | **3.61**% | 1.4% | −0.23% |
| PigArtPressure | −71.63% | 0.0% | −24.04% | **6.73**% |
| PowerCons | 0.0% | **3.89**% | −1.67% | 0.0% |
| ProximalPhalanxOutlineAgeGroup | −0.98% | 0.0% | 0.0% | **1.95**% |
| ProximalPhalanxOutlineCorrect | 1.03% | −0.34% | −0.34% | **2.06**% |
| ProximalPhalanxTW | −0.49% | **1.46**% | −1.95% | 0.49% |
| RefrigerationDevices | 0.27% | −0.53% | −0.8% | **1.6**% |
| Rock | −24.0% | −24.0% | **−6.0**% | −8.0% |
| ScreenType | **−0.27**% | −4.27% | −1.07% | −1.07% |
| SemgHandGenderCh2 | −9.67% | **2.33**% | 0.33% | 1.17% |
| SemgHandMovementCh2 | −16.0% | **11.33**% | 2.22% | 2.44% |
| SemgHandSubjectCh2 | −19.33% | **6.22**% | 2.0% | 2.43% |
| ShapeletSim | 1.67% | **10.56**% | 0.0% | 0.56% |
| ShapesAll | −12.0% | **−5.17**% | −8.33% | −0.33% |
| SmallKitchenAppliances | 4.0% | 1.07% | −1.33% | **4.2**% |
| SmoothSubspace | **0.0**% | −3.33% | −2.0% | 0.0% |
| SonyAIBORobotSurface1 | 2.16% | −5.99% | **2.66**% | 1.5% |
| SonyAIBORobotSurface2 | −1.99% | −4.09% | **−1.26**% | −1.24% |
| StarLightCurves | 0.22% | −0.34% | −1.07% | **1.27**% |
| Strawberry | −0.27% | −2.43% | −1.35% | **0.54**% |
| SwedishLeaf | 0.48% | **1.12**% | −6.4% | 0.33% |
| Symbols | −7.74% | −1.01% | **3.12**% | 2.23% |
| SyntheticControl | .0% | −1.33% | −0.33% | **1.0**% |
| ToeSegmentation1 | −1.75% | −0.44% | −6.58% | **3.07**% |
| ToeSegmentation2 | −1.54% | −6.92% | **2.31**% | 1.54% |
| Trace | **0.0**% | **0.0**% | −2.0% | **0.0**% |
| TwoLeadECG | **4.65**% | 0.0% | 3.78% | 2.81% |
| TwoPatterns | −2.2% | **−0.98**% | −1.6% | −1.63% |
| UMD | −2.78% | 0.0% | −2.78% | **4.29**% |
| UWaveGestureLibraryAll | −7.68% | −3.8% | −3.49% | **−0.87**% |
| UWaveGestureLibraryX | −3.6% | −1.48% | −2.18% | **0.87**% |
| UWaveGestureLibraryY | −1.12% | −6.0% | −2.07% | **0.34**% |
| UWaveGestureLibraryZ | −2.54% | **−0.5**% | −1.68% | −0.2% |
| Wafer | −0.15% | **0.15**% | −0.21% | −0.15% |
| Wine | −9.26% | 1.85% | **7.41**% | −1.85% |
| WordSynonyms | −6.58% | −6.74% | 0.0% | **0.63**% |
| Worms | −9.09% | −2.6% | −2.6% | **9.09**% |
| WormsTwoClass | −2.6% | 1.3% | **2.6**% | 1.3% |
| Yoga | −7.97% | −5.13% | −0.9% | **0.1**% |

## C  ENLARGED HYPERPARAMETERS SENSITIVITY ANALYSIS

Figures 8 to 17 show 3D plots of classifier performance as a function of $\alpha$ and the number of iterations for ASCENSION$_{\text{EmbCl}}$, FCN, and ResNet, across representative datasets from each category of the UCR archive. The name of each category and their representative datasets are detailed in Table 4.

$\alpha$ **parameter:**  As discussed in section 3.2.2, performance improvement relation to $\alpha$ seems difficult to generalize while remaining relatively stable. Increasing $\alpha$ can lead to better boundary exploration, as shown in Figures 12 and 11 but can also make the performance drop for too high values of $\alpha$. While pinpointing the exact $\alpha$ values and iterations for optimal results across all datasets is not trivial, the general trend suggests selecting $\alpha \in [1, 3]$ to expand class boundaries without venturing into areas that risk class overlap, which could negatively impact classification accuracy.

**Number of iterations:**  In Figures 11-13, and 15, we observe that a higher number of iterations can have either a positive or negative impact on performance, whereas in Figure 8, the number of iterations does not play a significant role in performance improvement. This ambivalent behavior is closely related to the class distribution within the dataset. As the number of iterations increases, classes in the latent space may become closer due to the increase in the $\alpha$ parameter at each iteration, which leads to the expansion of covariances $\alpha\Sigma_k$ (cf., section 2.2). Therefore, we recommend carefully adjusting the number of iterations in relation to the chosen $\alpha$ parameter.

Table 7: **Synthetic sample diversity** for ASCENSION, TTS-GAN, and FAA; results being aggregated based on the UCR benchmark dataset categories (cf., Table 4)

| Dataset | ASCENSION | TTSGAN | FAA |
|---|---|---|---|
| Worms | 28982.94 | 3710.44 | 845.62 |
| UMD | 174.16 | 319.86 | 52.14 |
| ECG200 | 186.58 | 69.09 | 57.18 |
| SemgHandMovementCh2 | $4.35 \times 10^8$ | $4.35 \times 10^7$ | $4.49 \times 10^7$ |
| ACSF1 | 82826.34 | 24652.55 | 7040.46 |
| Car | 79.75 | 122.37 | 33.57 |
| BeetleFly | 2911.51 | 2057.51 | - |
| Adiac | 63.47 | 15.33 | 0.87 |
| Ham | 1538.55 | 92.29 | 613.41 |

Table 8: **Synthetic sample quality** for ASCENSION, TTS-GAN, and FAA; results being aggregated based on the UCR benchmark dataset categories (cf., Table 4).

| Dataset | ASCENSION | | |
|---|---|---|---|
| | $D_{train}$ | $D_{test}$ | Ratio |
| Worms | 1325.33 | 1294.99 | 0.98 |
| UMD | 62.09 | 59.67 | 0.96 |
| ECG200 | 75.83 | 76.28 | 1.01 |
| SemgHandMovementCh2 | 39987.03 | 40152.58 | 1.00 |
| ACSF1 | 1082.69 | 1061.81 | 0.98 |
| Car | 74.99 | 75.14 | 1.00 |
| BeetleFly | 401.44 | 424.44 | 1.06 |
| Adiac | 28.89 | 29.22 | 1.01 |
| Ham | 301.30 | 302.32 | 1.00 |

| Dataset | TTS-GAN | | |
|---|---|---|---|
| | $D_{train}$ | $D_{test}$ | Ratio |
| Worms | 728.67 | 716.71 | 0.98 |
| UMD | 51.13 | 50.46 | 0.99 |
| ECG200 | 32.78 | 33.38 | 1.02 |
| SemgHandMovementCh2 | 18009.14 | 17619.67 | 0.98 |
| ACSF1 | 818.11 | 819.17 | 1.00 |
| Car | 130.32 | 131.80 | 1.01 |
| BeetleFly | 385.27 | 395.06 | 1.03 |
| Adiac | 39.26 | 39.35 | 1.00 |
| Ham | 125.93 | 131.80 | 1.05 |

| Dataset | FAA | | |
|---|---|---|---|
| | $D_{train}$ | $D_{test}$ | Ratio |
| Worms | 957.46 | 965.85 | 1.01 |
| UMD | 70.34 | 68.61 | 0.98 |
| ECG200 | 76.44 | 76.60 | 1.00 |
| SemgHandMovementCh2 | 17456.18 | 17200.74 | 0.99 |
| ACSF1 | 1061.23 | 1063.90 | 1.00 |
| Car | 665.28 | 665.07 | 1.00 |
| BeetleFly | 418.88 | 415.42 | 0.99 |
| Adiac | 184.91 | 185.01 | 1.00 |
| Ham | 323.65 | 320.79 | 0.99 |

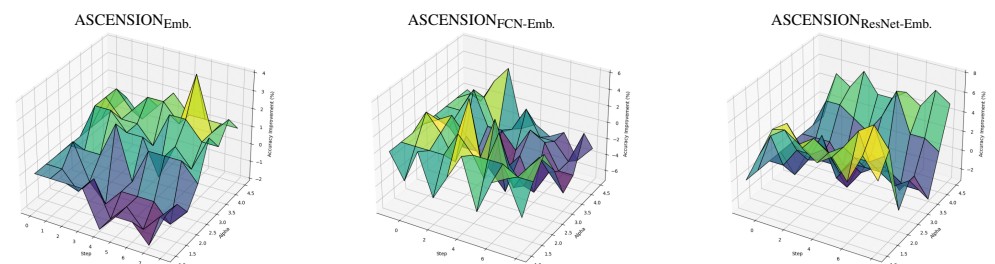

Figure 8: **ECG:** Classifier performance against $\alpha$ and iteration number for **ECG200** dataset.

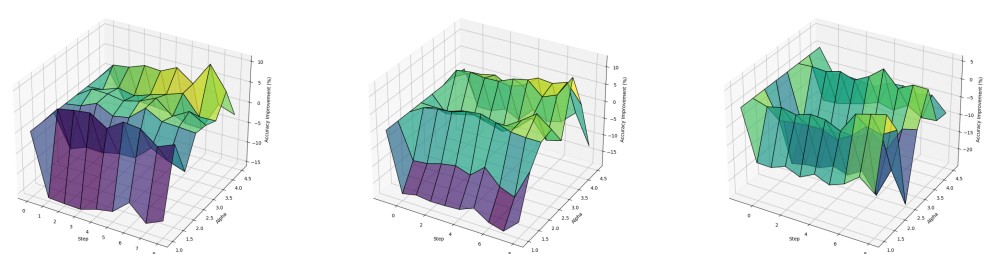

Figure 9: **EOG:** Classifier performance against $\alpha$ and iteration number for **EOGVerticalSignal**.

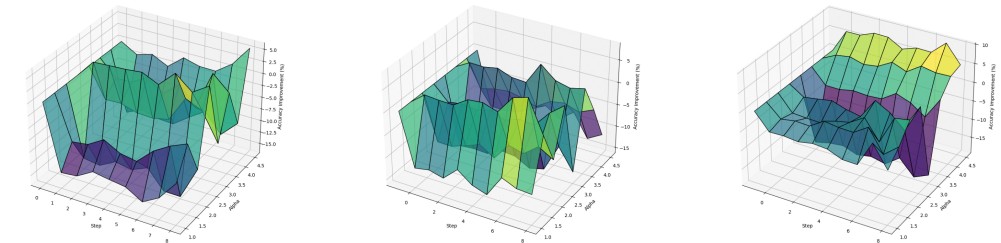

Figure 10: **Hemodynamics:** Classifier performance against $\alpha$ and iterations for **PigArtPressure**.

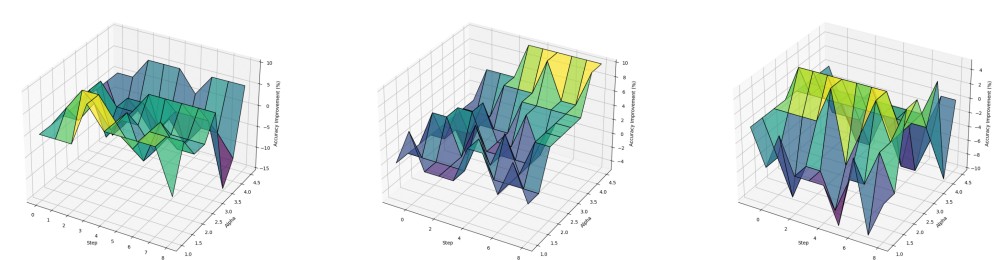

Figure 11: **Image:** Classifier performance against $\alpha$ and iteration number for **BeetleFly** dataset.

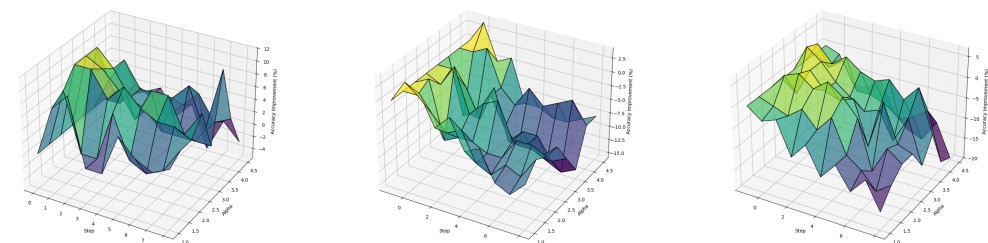

Figure 12: **Motion:** Classifier performance against $\alpha$ and iteration number for **Worms** dataset.

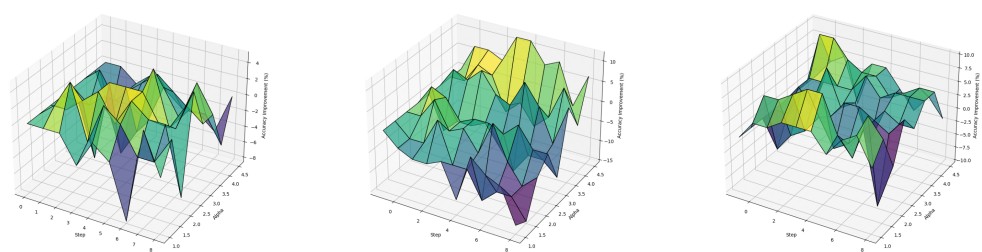

Figure 13: **Sensor:** Classifier performance against $\alpha$ and iteration number for **Car** dataset.

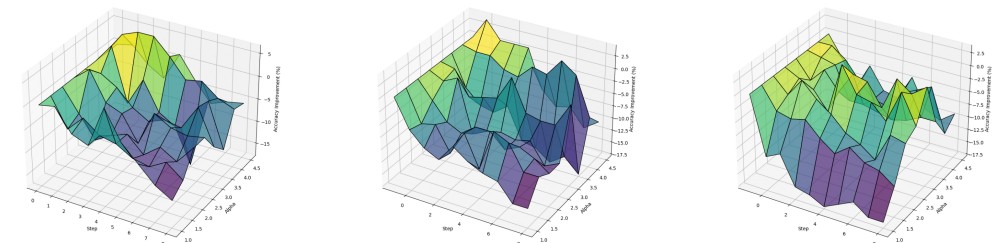

Figure 14: **Simulated:** Classifier performance against $\alpha$ and iteration number for **UMD** dataset.

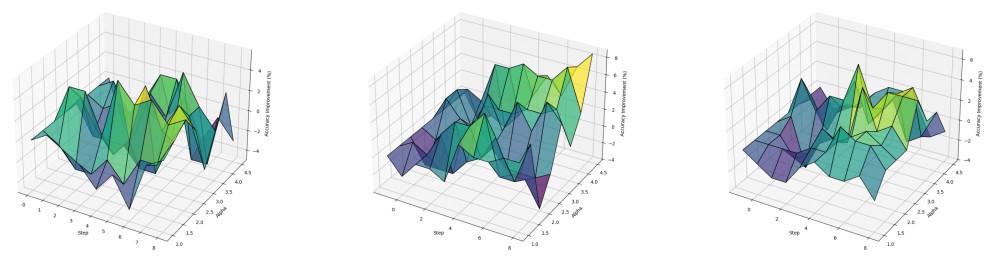

Figure 15: **Spectro:** Classifier performance against $\alpha$ and iteration number for **Ham** dataset.

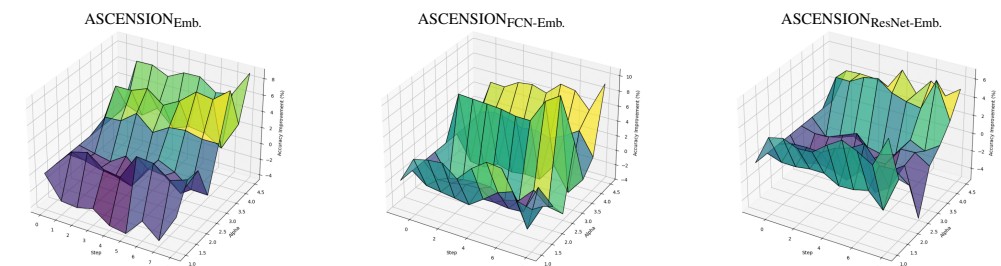

Figure 16: **Spectrum:** Classifier performance against $\alpha$ and iteration number for **SemgHandMovementCh2** dataset.

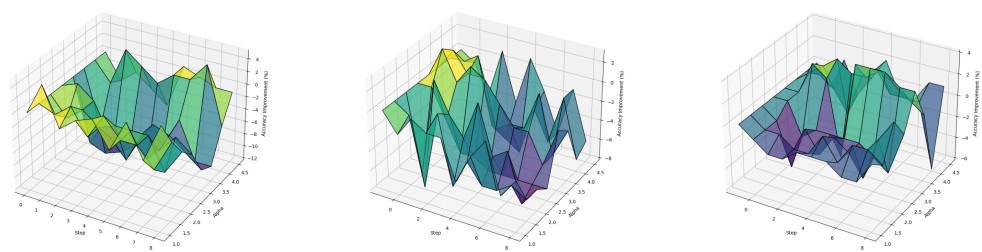

Figure 17: **Device:** Classifier performance against $\alpha$ and iteration number for **ACSF1** dataset.

## D    TIME SERIES FEATURES

In this section, we describe the 22 time series features (Catch22) presented in (Lubba et al., 2019), and the two additional features (denoted by F23 and F24 below) considered in this study.

**F1: `DN_HistogramMode_5`** Top z-score range based on the highest count from a 5-bin histogram, representing the most frequent distribution range in the dataset.

**F2: `DN_HistogramMode_10`** Similar to DN5, but this considers the top z-score range based on a 10-bin histogram, providing a finer resolution.

**F3: `CO_f1ecac`** Represents the first 1/e crossing of the autocorrelation function, indicating how quickly the autocorrelation of a time series decays.

**F4: `CO_FirstMin_ac`** Identifies the first minimum of the autocorrelation function, which helps analyze the periodicity of the time series.

**F5: `CO_HistogramAMI_even_2_5`** Automutual information for $m = 2$ and $\tau = 5$, capturing the dependency between data points across time.

**F6: `CO_trev_1_num`** This statistic measures time-reversibility, focusing on the differences between successive points in the time series raised to the third power.

**F7: `MD_hrv_classic_pnn40`** Proportion of successive differences in time series values that exceed 0.04 of the standard deviation, indicating rapid fluctuations.

**F8: `SB_BinaryStats_mean_longstretch1`** The longest period where values stay consecutively above the mean, representing persistent trends in the data.

**F9: `SB_TransitionMatrix_3ac_sumdiagcov`** Trace of the covariance of the transition matrix between symbols in a 3-letter alphabet, used to assess transitions in symbolized data.

**F10: `PD_PeriodicityWang_th0_01`** A periodicity measure, indicating how regularly patterns repeat within the time series.

**F11: `CO_Embed2_Dist_tau_d_expfit_meandiff`** Exponential fit to the differences in distances between successive points in a 2-dimensional embedding space, revealing structural relationships.

**F12: `IN_AutoMutualInfoStats_40_gaussian_fmmi`** First minimum of the automutual information function, which gives insight into the periodicity and structure of the time series.

**F13: `FC_LocalSimple_mean1_tauresrat`** Measures the change in correlation length after iteratively differencing the time series, providing insights into the stationarity of the data.

**F14: `DN_OutlierInclude_p_001_mdrmd`** Measures the time intervals between successive extreme events occurring above the mean, indicating patterns of high values.

**F15: `DN_OutlierInclude_n_001_mdrmd`** Similar to DNOp but for extreme events occurring below the mean, highlighting the time intervals between low-value outliers.

**F16: `SP_Summaries_welch_rect_area_5_1`** This computes the total power in the lowest fifth of the frequencies from a Fourier power spectrum, reflecting long-term trends.

**F17: `SB_BinaryStats_diff_longstretch0`** The longest period of successive decreases in the time series, capturing prolonged declining trends.

**F18: `SB_MotifThree_quantile_hh`** Shannon entropy of successive symbol pairs in a 3-letter quantile symbolization, quantifying the complexity of transitions between motifs.

**F19: `SC_FluctAnal_2_rsrangefit_50_1_logi_prop_r1`** Proportion of slower timescale fluctuations that scale with rescaled range fits, indicating long-term memory in the data.

**F20: `SC_FluctAnal_2_dfa_50_1_2_logi_prop_r1`** Proportion of slower timescale fluctuations that scale with detrended fluctuation analysis (DFA) under 50

**F21: `SP_Summaries_welch_rect_centroid`** The centroid of the Fourier power spectrum, which offers a measure of the central frequency or the dominant pattern in the time series.

**F22: `FC_LocalSimple_mean3_stderr`** Calculates the mean error from a rolling 3-sample mean forecast, capturing the volatility of short-term predictions.

**F23: `Train_Test_Ratio`** The ratio of training data to test data in the dataset.

**F24: `Discrepancy_in_Distance`** To estimate the discrepancy in distance between the training and testing set distributions, as defined in Appendix E.2

# E PERFORMANCE METRIC FORMALIZATION

## E.1 TRILEMMA METRICS

**Synthetic sample quality:** To quantify the quality of the generated samples, we compute the mean intra-class distance across all classes using Dynamic Time Warping (DTW) Senin (2008) as the distance metric. Let $\mathcal{Z}_k = z_{k,1}, z_{k,2}, \ldots, z_{k,n_k}$ represent the true data belonging to class $k$ and $\mathcal{X}_{gen,k} = x_{k,1}, x_{k,2}, \ldots, x_{k,n_k}$ represent the set of generated samples belonging to class $k$, and $q_k(X_k)$ the quality of synthetic sample set $X$ on class $k$:

$$ql_k(X_k) = \frac{1}{n_k} \sum_{i=1}^{n_k} \sum_{j=1}^{n_l} \text{DTW}(x_{k,i}, z_{k,j}) \tag{7}$$

We then express the quality $Q_{\text{method}}$ of a method on a dataset DS with $l$ class as :

$$QL_{\text{method}}(\text{DS}) = \frac{1}{l} \sum_{k=1}^{l} q_k(X_k) \tag{8}$$

**Diversity:** Let $DS$ be a dataset with $l$ classes, $\mathcal{Z}_k = z_{k,1}, z_{k,2}, \ldots, z_{k,n_k}$ represent the true data belonging to class $k$, $\mathcal{X}_{gen,k} = x_{k,1}, x_{k,2}, \ldots, x_{k,n_k}$ represent the set of generated samples belonging to class $k$, we can define the diversity $Div_{method}(\text{DS})$ as such :

$$Div_{method}(\text{DS}) = \frac{1}{l} \sum_{k=1}^{l} \text{Var}(\{\text{DTW}(x_k, \mu_k), x_k \in X_k\}) \tag{9}$$

where $\mu_k$ is the mean of the true samples in class $k$.

**Fast sampling:** GPU/hours is used.

### E.2 Discrepancy in distance between training and test sets

#### E.2.1 Formalization

To estimate the discrepancy in distance between the training and test sets, we compute the mean intra-class distance across all classes using DTW as the distance metric. Let $\mathcal{X}_k = x_{k,1}, x_{k,2}, \ldots, x_{k,n_k}$ represent the set of generated samples belonging to class $k$, and $d_k$ be the mean intra-class distance for class $k$, defined as:

$$d_k = \frac{1}{n_k} \sum_{i=1}^{n_k} \text{DTW}(x_{k,i}, \mu_k) \qquad (10)$$

where $\mu_k$ is the mean of the samples in class $k$ (computed using DTW barycenter averaging, where applicable). The overall dispersion $D$ of the dataset is then defined as the mean intra-class variance across all $K$ classes:

$$D = \frac{1}{K} \sum_{k=1}^{K} d_k \qquad (11)$$

To estimate the discrepancy between the training and test datasets, we compute the ratio between the dispersion of the test set $D_{\text{test}}$ and the diversity of the train set $D_{\text{train}}$. This ratio $V$ is defined as:

$$V = \frac{D_{\text{test}}}{D_{\text{train}}} \qquad (12)$$

The discrepancies ratio $V \approx 1$ indicates similar diversity between the train and test sets, while deviations from 1 suggest more diversity in the training set ($V < 1$) or in the test set ($V > 1$).

A dataset where the ratio $V > 1$ is considered to be more challenging for usual generative techniques, as the train set does not accurately represent the test set in these cases.
As such the datasets at the far right in

#### E.2.2 Experimental results

The discrepancy ratio of the 102 UCR datasets have been plotted in an ascending order in . Le us consider three datasets with extreme ratios: **(i) Discrepancy toward test:** Dataset `Car` (1.51); **(ii) No discrepancy:** Dataset `ECGFiveDays` (1.01); **(iii) Discrepancy toward train:** Dataset `EOGVerticalSignal` (0.77). Referring to the performance results in Table 6, we observe that ASCENSION consistently improves the classification performance, while TTS, FAA, and LA each reduce the classifier performance at least once.

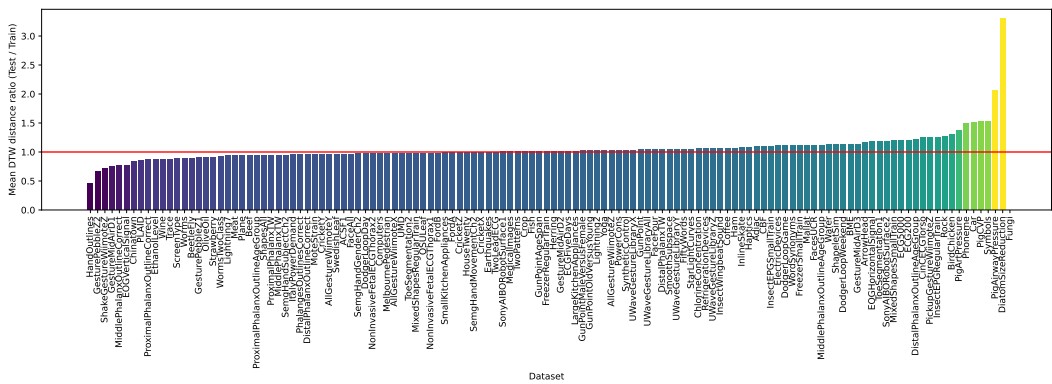

Figure 18: **Distribution discrepancy ratio:** Overview of the difference in discrepancy between training and testing sets of the 102 UCR datasets; discrepancy ratio computed using (12)
.

Detailed results of the discrepancies across datasets are available in Table 9

Table 9: Discrepancy Metrics Across Datasets

| Dataset | Ratio | Dispersion$_{TEST}$ | Dispersion$_{TRAIN}$ |
|---|---|---|---|
| HandOutlines | 0.46 | $1.50 \times 10^2$ | $1.39 \times 10^2$ |
| GesturePebbleZ2 | 0.66 | $3.09 \times 10^1$ | $3.02 \times 10^1$ |
| ShakeGestureWiimoteZ | 0.71 | $5.36 \times 10^2$ | $6.04 \times 10^2$ |
| GestureMidAirD1 | 0.75 | $4.18 \times 10^2$ | $4.30 \times 10^2$ |
| MiddlePhalanxOutlineCorrect | 0.77 | $1.01 \times 10^6$ | $1.02 \times 10^6$ |
| EOGVerticalSignal | 0.77 | $6.38 \times 10^3$ | $5.62 \times 10^3$ |
| Chinatown | 0.84 | $1.71 \times 10^3$ | $2.05 \times 10^3$ |
| PLAID | 0.85 | $3.50 \times 10^2$ | $3.38 \times 10^2$ |
| ProximalPhalanxOutlineCorrect | 0.87 | $1.34 \times 10^1$ | $1.48 \times 10^1$ |
| EthanolLevel | 0.87 | $3.18 \times 10^1$ | $2.10 \times 10^1$ |
| Wine | 0.87 | $3.34 \times 10^4$ | $3.33 \times 10^4$ |
| Trace | 0.88 | $4.46 \times 10^3$ | $4.41 \times 10^3$ |
| ScreenType | 0.88 | $2.18 \times 10^2$ | $2.46 \times 10^2$ |
| Worms | 0.89 | $1.13 \times 10^2$ | $1.00 \times 10^2$ |
| BeetleFly | 0.89 | $5.79 \times 10^1$ | $5.30 \times 10^1$ |
| GesturePebbleZ1 | 0.90 | $4.34 \times 10^0$ | $3.98 \times 10^0$ |
| OliveOil | 0.91 | $5.64 \times 10^0$ | $5.94 \times 10^0$ |
| Strawberry | 0.91 | $1.59 \times 10^2$ | $1.56 \times 10^2$ |
| WormsTwoClass | 0.93 | $4.09 \times 10^1$ | $4.26 \times 10^1$ |
| Lightning7 | 0.94 | $3.32 \times 10^1$ | $3.80 \times 10^1$ |
| Meat | 0.94 | $2.80 \times 10^3$ | $1.35 \times 10^3$ |
| Plane | 0.94 | $9.58 \times 10^1$ | $1.01 \times 10^2$ |
| Beef | 0.94 | $6.40 \times 10^1$ | $6.78 \times 10^1$ |
| ProximalPhalanxOutlineAgeGroup | 0.94 | $4.70 \times 10^2$ | $7.09 \times 10^2$ |
| ShapesAll | 0.94 | $4.40 \times 10^1$ | $3.95 \times 10^1$ |
| ProximalPhalanxTW | 0.94 | $1.39 \times 10^4$ | $1.36 \times 10^4$ |
| MiddlePhalanxTW | 0.94 | $4.74 \times 10^0$ | $5.02 \times 10^0$ |
| SemgHandSubjectCh2 | 0.95 | $5.14 \times 10^1$ | $5.28 \times 10^1$ |
| ItalyPowerDemand | 0.95 | $2.75 \times 10^0$ | $2.92 \times 10^0$ |
| PhalangesOutlinesCorrect | 0.95 | $2.02 \times 10^1$ | $2.00 \times 10^1$ |
| DistalPhalanxOutlineCorrect | 0.96 | $5.31 \times 10^0$ | $6.94 \times 10^0$ |
| MoteStrain | 0.96 | $3.27 \times 10^1$ | $2.60 \times 10^1$ |
| CricketY | 0.96 | $3.90 \times 10^2$ | $3.94 \times 10^2$ |
| AllGestureWiimoteY | 0.96 | $1.57 \times 10^1$ | $1.63 \times 10^1$ |
| SwedishLeaf | 0.96 | $4.69 \times 10^2$ | $4.37 \times 10^2$ |
| ACSF1 | 0.96 | $1.01 \times 10^3$ | $1.04 \times 10^3$ |
| FaceAll | 0.97 | $3.58 \times 10^1$ | $3.67 \times 10^1$ |
| SemgHandGenderCh2 | 0.97 | $1.47 \times 10^2$ | $1.53 \times 10^2$ |
| DodgerLoopDay | 0.97 | $6.13 \times 10^2$ | $6.62 \times 10^2$ |
| NonInvasiveFetalECGThorax2 | 0.97 | $2.52 \times 10^0$ | $2.42 \times 10^0$ |
| Computers | 0.97 | $1.94 \times 10^2$ | $1.98 \times 10^2$ |
| MelbournePedestrian | 0.97 | $7.90 \times 10^1$ | $7.41 \times 10^1$ |
| AllGestureWiimoteX | 0.97 | $1.63 \times 10^2$ | $1.64 \times 10^2$ |
| UMD | 0.97 | $1.89 \times 10^1$ | $1.89 \times 10^1$ |
| ToeSegmentation2 | 0.97 | $2.03 \times 10^2$ | $1.72 \times 10^2$ |
| MixedShapesRegularTrain | 0.98 | $4.20 \times 10^2$ | $4.76 \times 10^2$ |
| OSULeaf | 0.98 | $8.85 \times 10^3$ | $6.43 \times 10^3$ |
| NonInvasiveFetalECGThorax1 | 0.98 | $1.31 \times 10^2$ | $1.33 \times 10^2$ |
| FordB | 0.98 | $2.81 \times 10^0$ | $2.80 \times 10^0$ |
| SmallKitchenAppliances | 0.99 | $2.49 \times 10^1$ | $2.61 \times 10^1$ |

| Dataset | Ratio | Dispersion$_{TEST}$ | Dispersion$_{TRAIN}$ |
|---|---|---|---|
| FordA | 0.99 | $3.73 \times 10^3$ | $3.83 \times 10^3$ |
| CricketZ | 0.99 | $2.55 \times 10^1$ | $2.52 \times 10^1$ |
| HouseTwenty | 0.99 | $2.44 \times 10^0$ | $2.79 \times 10^0$ |
| SemgHandMovementCh2 | 1.00 | $1.23 \times 10^4$ | $1.24 \times 10^4$ |
| CricketX | 1.00 | $6.78 \times 10^1$ | $6.10 \times 10^1$ |
| Earthquakes | 1.00 | $1.31 \times 10^2$ | $1.24 \times 10^2$ |
| TwoLeadECG | 1.00 | $2.28 \times 10^1$ | $2.32 \times 10^1$ |
| SonyAIBORobotSurface1 | 1.00 | $8.36 \times 10^0$ | $8.36 \times 10^0$ |
| MedicalImages | 1.00 | $7.57 \times 10^1$ | $8.10 \times 10^1$ |
| TwoPatterns | 1.00 | $5.83 \times 10^2$ | $3.90 \times 10^2$ |
| Crop | 1.00 | $1.28 \times 10^4$ | $1.35 \times 10^4$ |
| Fish | 1.00 | $1.13 \times 10^3$ | $9.94 \times 10^2$ |
| GunPointAgeSpan | 1.00 | $5.50 \times 10^0$ | $4.90 \times 10^0$ |
| FreezerRegularTrain | 1.01 | $2.47 \times 10^3$ | $3.27 \times 10^3$ |
| Herring | 1.01 | $1.02 \times 10^1$ | $1.07 \times 10^1$ |
| GestureMidAirD2 | 1.01 | $6.39 \times 10^0$ | $6.13 \times 10^0$ |
| ECGFiveDays | 1.01 | $5.42 \times 10^1$ | $4.85 \times 10^1$ |
| LargeKitchenAppliances | 1.01 | $3.68 \times 10^1$ | $3.08 \times 10^1$ |
| GunPointMaleVersusFemale | 1.02 | $3.69 \times 10^1$ | $5.17 \times 10^1$ |
| GunPointOldVersusYoung | 1.02 | $5.70 \times 10^2$ | $6.35 \times 10^2$ |
| Lightning2 | 1.02 | $5.96 \times 10^1$ | $1.31 \times 10^2$ |
| Yoga | 1.02 | $3.02 \times 10^4$ | $2.97 \times 10^4$ |
| AllGestureWiimoteZ | 1.02 | $1.06 \times 10^1$ | $9.93 \times 10^0$ |
| PowerCons | 1.02 | $2.07 \times 10^4$ | $1.63 \times 10^4$ |
| SyntheticControl | 1.02 | $2.29 \times 10^2$ | $1.92 \times 10^2$ |
| UWaveGestureLibraryX | 1.02 | $6.81 \times 10^1$ | $6.67 \times 10^1$ |
| GunPoint | 1.04 | $3.83 \times 10^2$ | $3.91 \times 10^2$ |
| UWaveGestureLibraryAll | 1.04 | $5.73 \times 10^1$ | $5.46 \times 10^1$ |
| FaceFour | 1.04 | $5.44 \times 10^1$ | $5.14 \times 10^1$ |
| DistalPhalanxTW | 1.04 | $2.07 \times 10^1$ | $2.07 \times 10^1$ |
| SmoothSubspace | 1.04 | $4.86 \times 10^1$ | $3.19 \times 10^1$ |
| UWaveGestureLibraryY | 1.05 | $2.00 \times 10^1$ | $1.73 \times 10^1$ |
| FiftyWords | 1.05 | $3.80 \times 10^0$ | $4.03 \times 10^0$ |
| StarLightCurves | 1.05 | $5.40 \times 10^4$ | $4.59 \times 10^4$ |
| ChlorineConcentration | 1.05 | $9.02 \times 10^1$ | $9.00 \times 10^1$ |
| RefrigerationDevices | 1.05 | $4.23 \times 10^1$ | $4.01 \times 10^1$ |
| UWaveGestureLibraryZ | 1.06 | $8.64 \times 10^0$ | $9.18 \times 10^0$ |
| InsectWingbeatSound | 1.06 | $7.54 \times 10^2$ | $7.85 \times 10^2$ |
| Coffee | 1.07 | $8.05 \times 10^0$ | $8.45 \times 10^0$ |
| Ham | 1.07 | $4.23 \times 10^2$ | $3.75 \times 10^2$ |
| InlineSkate | 1.07 | $8.25 \times 10^0$ | $6.80 \times 10^0$ |
| Haptics | 1.08 | $3.27 \times 10^1$ | $2.98 \times 10^1$ |
| Adiac | 1.09 | $2.81 \times 10^1$ | $2.25 \times 10^1$ |
| CBF | 1.09 | $6.69 \times 10^4$ | $8.63 \times 10^4$ |
| InsectEPGSmallTrain | 1.10 | $1.63 \times 10^2$ | $1.64 \times 10^2$ |
| ElectricDevices | 1.10 | $1.02 \times 10^2$ | $9.84 \times 10^1$ |
| DodgerLoopGame | 1.10 | $6.43 \times 10^2$ | $6.10 \times 10^2$ |
| WordSynonyms | 1.11 | $4.32 \times 10^3$ | $5.08 \times 10^3$ |
| FreezerSmallTrain | 1.11 | $2.29 \times 10^2$ | $2.35 \times 10^2$ |
| Mallat | 1.11 | $2.40 \times 10^1$ | $2.32 \times 10^1$ |

| Dataset | Ratio | Dispersion$_{TEST}$ | Dispersion$_{TRAIN}$ |
|---|---|---|---|
| FacesUCR | 1.12 | $1.20 \times 10^3$ | $1.08 \times 10^3$ |
| MiddlePhalanxOutlineAgeGroup | 1.12 | $2.70 \times 10^1$ | $2.24 \times 10^1$ |
| Wafer | 1.12 | $2.24 \times 10^2$ | $2.30 \times 10^2$ |
| ShapeletSim | 1.14 | $1.41 \times 10^4$ | $1.46 \times 10^4$ |
| ArrowHead | 1.16 | $1.71 \times 10^0$ | $1.88 \times 10^0$ |
| EOGHorizontalSignal | 1.18 | $3.01 \times 10^1$ | $2.65 \times 10^1$ |
| ToeSegmentation1 | 1.18 | $2.19 \times 10^2$ | $2.16 \times 10^2$ |
| SonyAIBORobotSurface2 | 1.18 | $2.80 \times 10^1$ | $2.36 \times 10^1$ |
| MixedShapesSmallTrain | 1.19 | $1.59 \times 10^2$ | $1.55 \times 10^2$ |
| ECG5000 | 1.19 | $4.17 \times 10^1$ | $4.77 \times 10^1$ |
| ECG200 | 1.21 | $1.28 \times 10^2$ | $1.25 \times 10^2$ |
| DistalPhalanxOutlineAgeGroup | 1.21 | $6.78 \times 10^1$ | $6.71 \times 10^1$ |
| CinCECGTorso | 1.24 | $1.41 \times 10^1$ | $1.40 \times 10^1$ |
| PickupGestureWiimoteZ | 1.25 | $5.23 \times 10^0$ | $5.98 \times 10^0$ |
| InsectEPGRegularTrain | 1.26 | $1.88 \times 10^1$ | $1.94 \times 10^1$ |
| Rock | 1.27 | $1.16 \times 10^2$ | $1.11 \times 10^2$ |
| BirdChicken | 1.30 | $5.28 \times 10^1$ | $5.47 \times 10^1$ |
| PigArtPressure | 1.38 | $1.03 \times 10^2$ | $9.85 \times 10^1$ |
| Phoneme | 1.50 | $5.18 \times 10^1$ | $4.70 \times 10^1$ |
| Car | 1.51 | $3.94 \times 10^2$ | $3.95 \times 10^2$ |
| PigCVP | 1.52 | $6.68 \times 10^1$ | $6.54 \times 10^1$ |
| Symbols | 1.53 | $1.23 \times 10^1$ | $3.72 \times 10^0$ |
| PigAirwayPressure | 2.07 | $7.11 \times 10^2$ | $5.72 \times 10^2$ |
| DiatomSizeReduction | 3.30 | $1.52 \times 10^3$ | $1.00 \times 10^3$ |

## F  EVOLUTION OF LATENT SPACE THROUGH LEARNING PHASE

A progressive visualization of the latent space offers valuable insights into the evolving distribution modeling and exploration process. Initially, the latent space representations exhibit fine clustering, but as we iterate in the augmentation loop, the latent space distributions become denser, enhancing the exploration part of these distributions. However, in the later stages of augmentation, the exploration process becomes increasingly challenging as the inter-class distances appear to shrink due to prior augmentation steps. It is important to note that these visualizations provide only a limited view of the actual distributions, as they are restricted to three dimensions (from an original 50-dimensional space).

Table 10: Latent Space Evolution. Visualization of the latent space for the 3 first dimensions (out of 50)

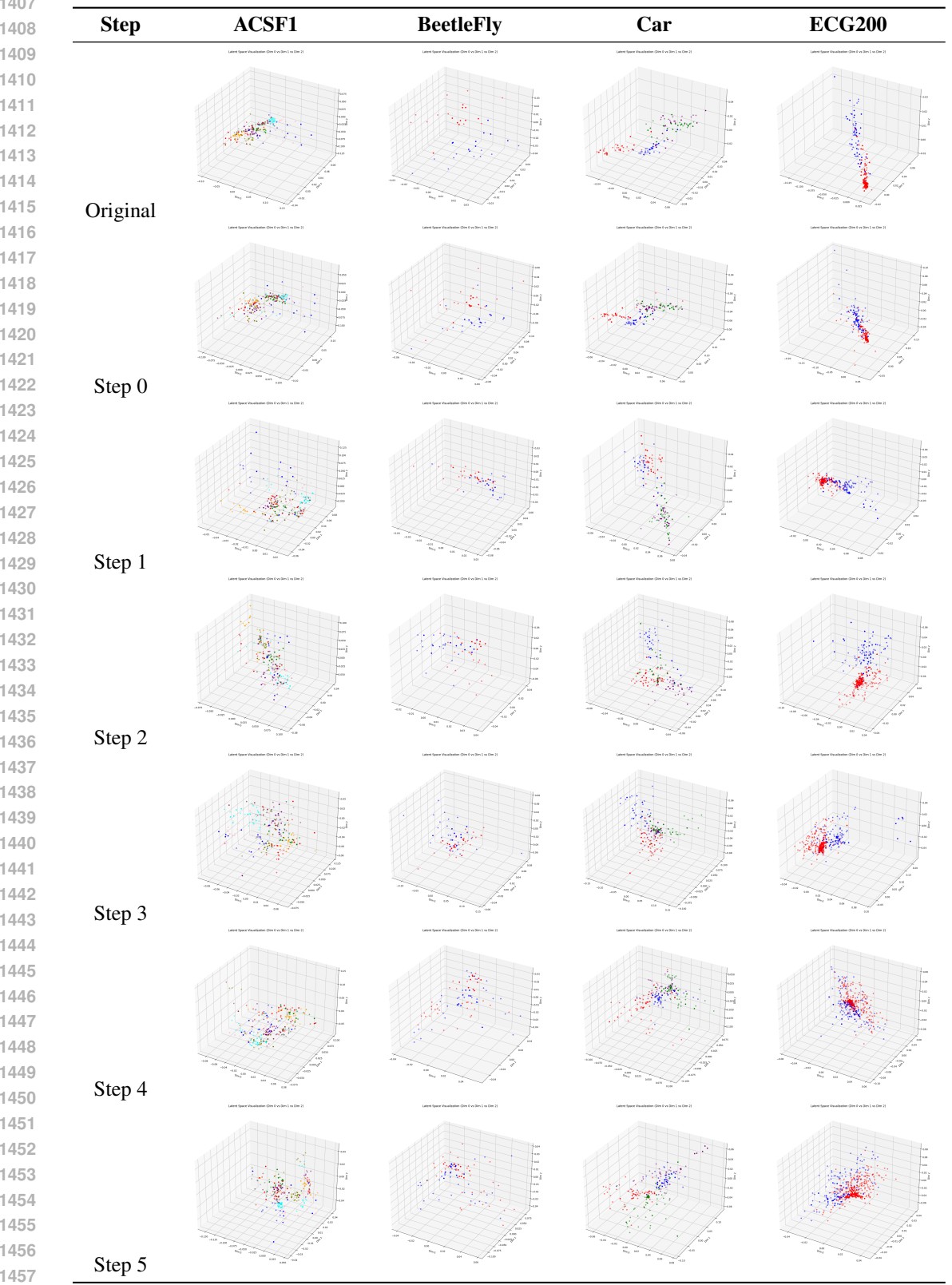

| Step | EOGVerticalSignal | Ham | PigArtPressure | SemgHandMov.Ch2 |
|------|-------------------|-----|----------------|-----------------|

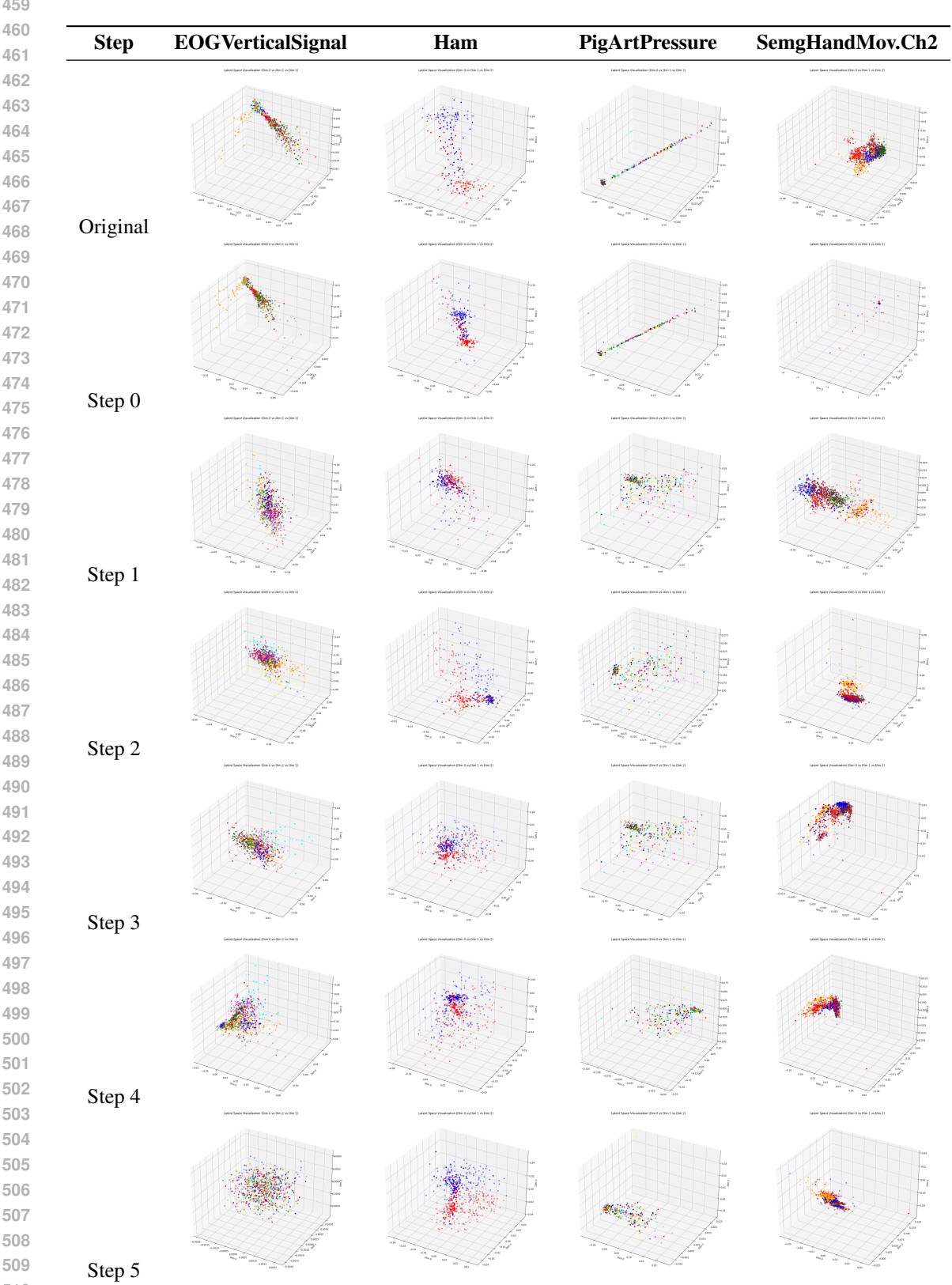

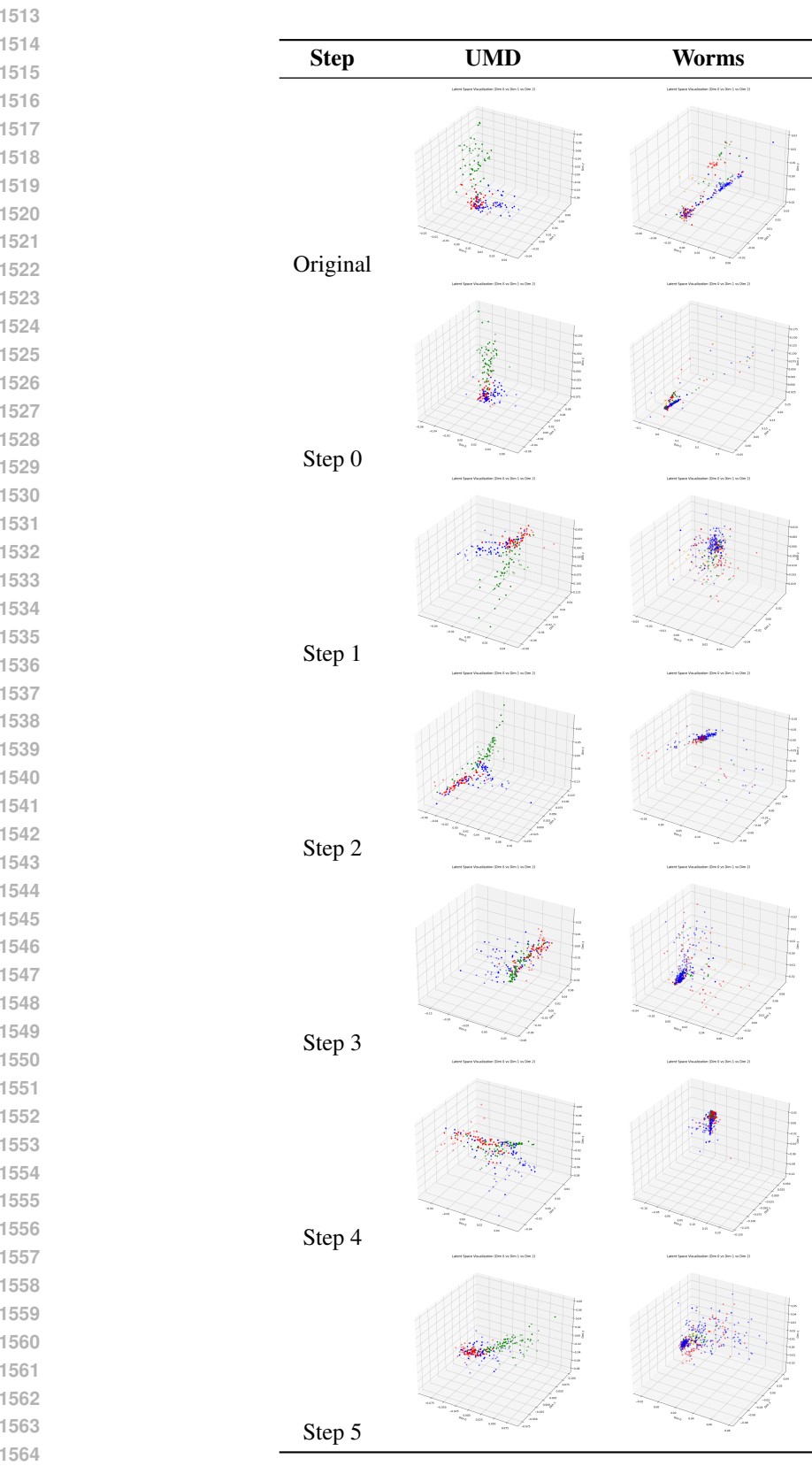

