# OpenReview forum: "Tackling the Generative learning trilemma through VAE and GMM-controlled latent space class expansion"
_ICLR.cc/2025/Conference — Submitted to ICLR 2025_

### Official Review · Reviewer_D18L · 2024-10-29

**Soundness:** 2
**Presentation:** 2
**Contribution:** 2
**Rating:** 3
**Confidence:** 3

**Summary:**

The paper proposes a data augmentation strategy for time series to address what the authors refer to as the generative learning trilemma, i.e. achieving high-quality generations, diversity in samples, and fast sampling at the same time. The proposed strategy consists in training a VAE model with clustering and classification terms added to the loss, followed by the training of a GMM model on the latent representations. Finally the variance hyperparameter of each of the components of the GMM model is scaled by a factor $\alpha$ to produce samples that arguably expand the training set boundary. The proposed data augmentation technique is compared with alternative approaches on the UCR Time Series Archive, evaluating its impact on time series classifiers performance.

**Strengths:**

- The paper deals with the relevant problem of data augmentations for time series.
- The experimental results show superior performance compared to the reported baselines, and interesting trends that indicate the proposed method positively impacts generalization of time series classifiers.
- The authors make an effort to provide ablations for sensitivity to hyperparameters, and explain negative performance of their method in selected datasets.

**Weaknesses:**

- Title and scope: This paper is specific to time series data, in particular to data augmentation for time series. However, neither the title nor the keywords make this explicit. I believe this should be adjusted, as the experiments and the baselines are limited to time series data. Note also that there is a previous paper titled *Tackling the Generative Learning Trilemma with Denoising Diffusion GANs* [1], where the first part of the title exatcly aligns with the title of this paper. Therefore I suggest to rewrite the title so that this is not the case.
- The paper proposes a two stage training approach using a VAE with added clustering and classification loss terms, followed by a GMM model fitted on the latent representations. Is there a reason why the authors did not directly train a VAE with a GMM prior (e.g. as in [2])?
- Not fully clear/convincing experiments, e.g.:
     - Essentially the results in Table 2 show that, at least on the presented UCR benchmark, none of the baseline DA methods gives an advantage, when looking at the full range of datasets. I do not have extensive expertise on DA strategies designed specifically for time series, but I find it hard to believe that this reflects the current state-of-the-art in the literature. Looking at these results I suspect there being relevant baselines in the literature that were not included.
     - I do not seem to find the information on the metric used to evaluate diversity in table 3.
     - Not sure the results in Figure 5 are clear to me. The authors train a RF classifier to predict the augmentation technique, by training on 24 features extracted and averaged for all datasets. Why are the feature importances specific to an augmentation method? Aren't these importance features relative to the classifier, and not the specific class (augmentation in this case).
- The paper is not easy to follow and certain details are hard to grasp, e.g:
     - Lines 207-210: what does "significant overlap" mean? is there a threshold?
     - Lines 235-237: what stated here seems to clash with that stated in Lines 207-210. If there is overlap, you look at the posterior distribution and reassign the label, but at the same time if a point sampled from a given class has higher density in another class it is discarded? Are these two strategies in place at the same time? Doesn't the second one make the first one useless?








[1] Xiao et al. Tackling the Generative Learning Trilemma with Denoising Diffusion GANs, ICLR, 2022.
[2] Jian et al. Variational Deep Embedding: An Unsupervised and Generative Approach to Clustering, arXiv:1611.05148, 2017.

**Questions:**

See weaknesses section.

---

### Official Review · Reviewer_9dW5 · 2024-11-01

**Soundness:** 2
**Presentation:** 2
**Contribution:** 2
**Rating:** 3
**Confidence:** 4

**Summary:**

In this paper, the authors tackle the generative learning trilemma. Specifically, the authors are trying to improve the generation sample quality of VAEs in time-series generation. To do so, the authors propose GMM-controlled latent space class expansion. The authors conduct their experiments on the UCR benchmark and evaluate the generation abilities through the improvement over downsteram classification tasks.

**Strengths:**

The author's method outperforms other data-augmentation methods for time series for downstream classification tasks on multiple datasets.

**Weaknesses:**

- If the model itself is able to learn separation in its latent space between different classes, what is its performance in classifying the data without the need for augmentation? Since the data came from this latent space in the first place, why is it necessary to use another classifier for these samples? I would expect such an ablation study to appear in this work.
- In terms of novelty, the authors claim: “none have ever proposed and integrated such a mechanism into VAEs.” I wonder what the core difference is between the proposed work and previous work on VAEs and GMMs (other than the application domain), such as [1] and [2].
- Part of the related work is missing. For example, work on Diffusion generative models for time series data [3], work on GANs [4], and especially work on VAEs [5].
- The model is compared with DA methods that actually, on average, worsen the model's performance, which seems weird to me to compare with since not using any DA method at all will result in better results on average. I think that in this case, if there are no standardized methods that improve, it is to use previous methods (mentioned above) and implement a naive conditioning mechanism to generate samples guided by class conditions as done in images.


[1] "Semi-supervised Learning with Deep Generative Models". Diederik P. Kingma et al.

[2] "Gaussian Mixture Variational Autoencoder with Contrastive Learning for Multi-Label Classification". Junwen Bai et al.

[3] "Diffusion-TS: Interpretable Diffusion For General Time Series Generation". Xinyu Yuan and Yan Qiao

[4] "GT-GAN: General Purpose Time Series Synthesis with Generative Adversarial Networks". Jinsung Jeon et al.

[5] "Generative modeling of regular and irregular time series data via Koopman VAEs." Ilan Naiman et al.

**Questions:**

See above

---

### Official Review · Reviewer_uNXi · 2024-11-02

**Soundness:** 2
**Presentation:** 2
**Contribution:** 1
**Rating:** 3
**Confidence:** 4

**Summary:**

This paper introduces ASCENSION, a generative DA method that combines VAEs with GMMs to enhance time series classification. ASCENSION aims to tackle the generative learning trilemma—balancing high-quality samples, diversity, and fast sampling—by controlling latent space exploration and expanding class boundaries in cases with training-testing distribution discrepancies. Through empirical testing on the UCR benchmark dataset, the authors report significant classification improvements over existing methods.

**Strengths:**

The control of latent space exploration (structured sampling) and the expansion of class boundaries through ASCENSION to enhance class separability and minimize boundary overlap seems novel. Also, the unique integration of latent space structuring to enable more targeted data augmentation with iterative retraining distinguishes the work as a promising advancement in generative data augmentation for complex temporal data.

**Weaknesses:**

A notable weakness of this paper is its tendency to present certain techniques as novel contributions when, in fact, they are well-documented and widely used in existing literature. Elements such as latent space clustering and the use of GMMs for class separation are established methods [1], yet the authors present them as if they were unique to this work. Moreover, the genuinely novel aspects, such as the controlled latent space exploration and iterative retraining process, lack depth and mathematical rigor, leaving these contributions underdeveloped.

Hypothesis 1 which is about clustering constraints in latent spaces to achieve structured representation and improve class consistency is indeed a well-documented approach in the literature, and it is more of an established method than a hypothesis. If this "hypothesis" serves only as a premise or background for their work, it would be clearer if presented as a methodological foundation rather than as a hypothesis. Ideally, hypotheses in a technical paper like this would be grounded in mathematical definitions, with clear conditions and quantifiable expectations. Authors could cite relevant works of structuring the latent space such as [2].

Also. these "hypotheses" do read more like general remarks than rigorously defined testable hypotheses.

This sentence is grammatically incorrect "The VAE fVAE to learn a low-dimensional representation of the input time series data."

MY ISSUE is that the authors do not properly separate their contributions and develop them rigorously. Everything new in this paper is not substantially treated.

Van de Ven, Gido M., Hava T. Siegelmann, and Andreas S. Tolias. "Brain-inspired replay for continual learning with artificial neural networks." Nature communications 11.1 (2020): 4069.

Tong, Shengbang, et al. "Incremental learning of structured memory via closed-loop transcription." arXiv preprint arXiv:2202.05411 (2022).

**Questions:**

How would the authors theoretically justify the proposed latent space exploration to enhance class separability and minimize boundary overlap?

---

### Official Review · Reviewer_tYKQ · 2024-11-04

**Soundness:** 3
**Presentation:** 3
**Contribution:** 3
**Rating:** 6
**Confidence:** 3

**Summary:**

This manuscript introduces an innovative generative technique known as ASCENSION (VAE and GMM-controlled latent space class expansion). This method preserves the advantages of Variational Autoencoders (VAE) in terms of diversity and rapid sampling, while facilitating a controlled and measurable exploration of previously uncharted areas within the latent space. This strategy not only improves classification accuracy but also produces higher quality, more realistic samples. ASCENSION utilizes the probabilistic characteristics of the VAE's latent space to model classes as Gaussian mixture models (GMMs).

**Strengths:**

This paper is well-written and the motivation is good.

**Weaknesses:**

There are several weaknesses:

1. The English language should be improved.

2. The main idea seems not very novel. This paper should provide a strong motivation.

3. The experiment can be further improved by providing more results and analysis.

4. The main idea is not very new since combining two generative models in a unified framework has been proposed in other studies.

**Questions:**

Please see the weakness section.

---

### Meta-Review · Area_Chair_r8Mt · 2024-12-18

**Metareview:**

The paper presents a method for data augmentation used to improve discriminative classifiers. The method is based on a VAE trained with a GMM prior. To "expand" the space of the model, samples are drawn with increasing variance from each class' component distribution in the hopes of improving sample diversity. The paper shows improvement over baseline data augmentation methods. The key weaknesses of the work are its potential novelty (as the generative modeling methods used are very mature and have been applied to data augmentation many times in the past), lack of theoretical justification for the method, experimental concerns (such as the fact that baseline data augmentation seems to reduce model performance), and the clarify of the writing and explanation.

Reviewers generally recommended rejection and I will recommend the same.

**Additional Comments On Reviewer Discussion:**

Reviewers provided initial, negative leaning reviews with consistent feedback. Authors did not participate in the rebuttal process.

---

### Decision · Program_Chairs · 2025-01-22

Reject